EMBO
Molecular Medicine

# Targeting endothelin receptor signalling overcomes heterogeneity driven therapy failure

Michael P Smith[1], Emily J Rowling[1], Zsofia Miskolczi[1], Jennifer Ferguson[1], Loredana Spoerri[2], Nikolas K Haass[2,3], Olivia Sloss[1], Sophie McEntegart[1], Imanol Arozarena[1,4], Alex von Kriegsheim[5], Javier Rodriguez[5], Holly Brunton[1], Jivko Kmarashev[6], Mitchell P Levesque[6], Reinhard Dummer[6], Dennie T Frederick[7], Miles C Andrews[8], Zachary A Cooper[8], Keith T Flaherty[7], Jennifer A Wargo[8] & Claudia Wellbrock[1,*] [ID]

## Abstract

Approaches to prolong responses to BRAF targeting drugs in melanoma patients are challenged by phenotype heterogeneity. Melanomas of a "MITF-high" phenotype usually respond well to BRAF inhibitor therapy, but these melanomas also contain subpopulations of the *de novo* resistance "AXL-high" phenotype. > 50% of melanomas progress with enriched "AXL-high" populations, and because AXL is linked to de-differentiation and invasiveness avoiding an "AXL-high relapse" is desirable. We discovered that phenotype heterogeneity is supported during the response phase of BRAF inhibitor therapy due to MITF-induced expression of endothelin 1 (EDN1). EDN1 expression is enhanced in tumours of patients on treatment and confers drug resistance through ERK re-activation in a paracrine manner. Most importantly, EDN1 not only supports MITF-high populations through the endothelin receptor B (EDNRB), but also AXL-high populations through EDNRA, making it a master regulator of phenotype heterogeneity. Endothelin receptor antagonists suppress AXL-high-expressing cells and sensitize to BRAF inhibition, suggesting that targeting EDN1 signalling could improve BRAF inhibitor responses without selecting for AXL-high cells.

**Keywords** AXL; BRAF; endothelin; melanoma; MITF
**Subject Categories** Cancer; Skin

See also: **T Kuilman & DS Peeper** (August 2017)

## Introduction

The MAP-kinase (MAPK) pathway is deregulated in the majority of malignant melanomas, and targeting the primary driver of hyperactive MAPK signalling, BRAF shows impressive initial responses in patients. However, prolonged responses are challenged by the development of resistance, often through mechanisms that allow bypassing BRAF inhibition (Lito *et al*, 2013; Carlino *et al*, 2015). These mechanisms can in principle be overcome by combining BRAF with MEK inhibitors, and indeed, BRAF/MEK combination therapies show significant improvement in progression-free survival compared to BRAF inhibitor monotherapy (Flaherty *et al*, 2012; Larkin *et al*, 2014; Long *et al*, 2016). However, some melanomas, despite harbouring a mutant *BRAF* gene, express higher levels of additional oncogenic drivers that confer intrinsic MAPK inhibitor resistance. These melanomas are characterized by gene signatures, which correlate with enhanced expression of the receptor tyrosine kinase AXL (Sensi *et al*, 2011; O'Connell *et al*, 2013; Konieczkowski *et al*, 2014; Muller *et al*, 2014; Tirosh *et al*, 2016). Nevertheless, with objective response rates of 60–70%, the majority of patients with BRAF mutant melanoma respond to MAPK inhibitors, and if we are to improve progression-free survival in these patients, it is of paramount importance to understand the biology of the responding tumours before and on treatment.

Melanomas that regress with MAPK pathway inhibitors are characterized by the expression of the lineage-specific transcription factor MITF (Konieczkowski *et al*, 2014; Wellbrock & Arozarena, 2015). MITF is detected in ~70% of treatment naive melanomas, which have been classified as MITF-high (Sensi *et al*, 2011; Tirosh *et al*, 2016). However, the situation is more complex; while weak MITF-expressing cells are sensitive to MAPK inhibitors,

1   Manchester Cancer Research Centre, Faculty of Biology, Medicine and Health, The University of Manchester, Manchester, UK
2   Translational Research Institute, The University of Queensland Diamantina Institute, The University of Queensland, Brisbane, Qld, Australia
3   Discipline of Dermatology, University of Sydney, Sydney, NSW, Australia
4   Navarrabiomed-Fundación Miguel Servet-Idisna, Pamplona, Spain
5   Systems Biology Ireland, School of Medicine, UCD, Dublin 4, Ireland
6   Department of Dermatology, Universitätsspital Zürich, University of Zurich, Zurich, Switzerland
7   Department of Medicine, Massachusetts General Hospital Cancer Center, Boston, MA, USA
8   Division of Surgical Oncology, University of Texas MD Anderson Cancer Center, Houston, TX, USA
    *Corresponding author. Tel: +44 161 2755189; E-mail: claudia.wellbrock@manchester.ac.uk

up-expression of MITF provides resistance (Haq *et al*, 2013; Smith *et al*, 2013, 2016; Muller *et al*, 2014). This is reflected in the fact that some patients relapse with up-regulated MITF expression, partly due to gene amplification (Muller *et al*, 2014; Van Allen *et al*, 2014; Smith *et al*, 2016).

As mentioned above, at bulk-tumour level, melanomas can be classified as either MITF-high or AXL-high, but single-cell sequencing has revealed intra-tumour heterogeneity, whereby MITF-high melanomas also contain cells with elevated AXL expression (Tirosh *et al*, 2016). Importantly, this heterogeneity can have profound consequences for therapy response; when patients eventually acquire resistance to MAPK inhibitors, > 50% of resistant tumours show enrichment for AXL-high populations (Tirosh *et al*, 2016). As high AXL expression is linked to a more invasive phenotype (Sensi *et al*, 2011; Muller *et al*, 2014), acquiring resistance with this phenotype could lead to a more aggressive state, which could be further supported by the regressing tumour microenvironment (Obenauf *et al*, 2015). Thus, avoiding a relapse with AXL-high tumours would be desirable for the implementation of salvage therapies. With this in mind, and to improve our understanding of the complexity of MITF-high melanomas in the context of MAPK inhibitor resistance, we set out to analyse the dynamics of individual MITF-expressing subpopulations during treatment with BRAF inhibitor.

# Results

### Heterogeneous MITF expression is maintained during BRAF inhibitor treatment

We have shown recently that in ~80% of patients on treatment with MAPK inhibitors bulk-tumour MITF mRNA increases due to transcriptional up-regulation (Smith *et al*, 2016). Furthermore, we revealed that MITF up-expression enhances MAPK inhibitor resistance during the drug-induced tolerance phase preceding acquired resistance (Smith *et al*, 2016).

To assess the consequences of MITF up-expression within a tumour at the single-cell level, we analysed a "MITF-high" melanoma sample from a patient, who showed a response on vemurafenib treatment (BRAFi, Appendix Table S1 for patient information). We found that before treatment basal MITF expression was heterogeneous with pools of weak, strong or undetectable MITF-expressing cell populations (Fig 1A). This was seen also in other melanoma biopsies (Fig EV1A) and is entirely in line with single-cell analysis data from MITF-high melanomas (Tirosh *et al*, 2016). Confirming our previous findings, 2 weeks into treatment, bulk-tumour MITF expression was increased in the tumour of patient 24 (Fig 1A). However, at the individual cell level, the MITF expression pattern was still heterogeneous. A similar heterogeneity was seen in other tumour samples from patients on treatment, even when the overall expression level of MITF did not increase (Appendix Fig S1). This suggested that despite MITF's function in drug resistance, no stringent selection for cells with increased MITF expression levels had occurred on treatment. Nevertheless, the heterogeneity could be due to the presence of populations of cells that differ in their genetic background, enabling them to resist the drug insult independently of MITF.

To address this issue, we analysed melanoma xenografts derived from the MITF-expressing cell line A375 (MITF-high), and as such a

population of genetically identical cells. Again we found that while MITF was up-regulated in BRAF inhibitor-responding tumours, its expression was heterogeneous throughout, and strong and weak MITF-expressing cells were detectable (Fig EV1B). Possibly, in this *in vivo* situation stroma-derived signals from the local tumour microenvironment could have differing effects on MITF expression (Smith *et al*, 2014). We therefore isolated cells from A375 xenografts that responded to and regressed on BRAF inhibitor. The overall increase in MITF expression was still detectable in these *ex vivo* cultures in the absence of a microenvironment, but intriguingly MITF heterogeneity prevailed, and stronger and weaker MITF-expressing cells were detected (Fig 1B). Importantly, the presence of weaker MITF-expressing cells was not due to enrichment for a "AXL-high/MITF-low" population—considered the most resistant phenotype—as this fraction was rather reduced in cultures responding to BRAF inhibitor (Fig EV1C and D).

We therefore attempted to monitor the dynamics of individual cells within one MITF-high cell line in the response to MAPK inhibition in more detail. To identify a representative cell line, we assessed the AXL and MITF expression status in a panel of melanoma cell lines and their link to response to BRAF inhibition. In agreement with previous reports, we found a correlation with high AXL expression and low MITF expression and resistance to BRAF inhibition (Fig 1C). The group of MITF-expressing cell lines displayed a considerable distribution of MITF expression levels, and whereas weaker expression correlated with BRAF inhibitor sensitivity, increased MITF expression protected from BRAF inhibition (Fig 1C).

We chose WM164 cells as they express intermediate MITF and AXL levels and respond to BRAF inhibition (Fig 1C). In untreated WM164 cells, MITF expression is heterogeneous (Fig 1D), which allowed us to assess whether high MITF expression will be selected for over the time of treatment. Using the FUCCI system, which can report on the individual phases of the cell cycle, we followed single FUCCI-WM164 cells (Haass *et al*, 2014) over the course of treatment for 3 days, a time suitable for real-time imaging and during which we already observe MITF up-expression in response to BRAF inhibitor (Fig 1D). In a DMSO, control population cells cycle in a fairly asynchronous mode with up to three cell divisions over 72 h (Fig 1E). Treatment with BRAF inhibitor led to a G1 arrest within 12 h in the majority of cells. Within 24–48 h after the G1 arrest, ~20% of cells died (detectable by DRAQ7 staining) and another 10% started dying during the remaining time of the experiment (Fig 1E). Occasionally, we observed that after exiting mitosis one daughter cell died, while the other daughter cell stayed arrested in G1 (Fig 1E, dashed line), suggesting great complexity in inhibitor response even within one cell line. While overall the behaviour of WM164 cells is in line with selection for more drug-resistant cells arrested in G1, this was not reflected at the single-cell MITF expression level, where heterogeneity was maintained (Fig 1D). Again, there was no enrichment for AXL-high/MITF-low cells in BRAF inhibitor-responding WM164 cells (Fig EV1E and F).

### BRAF inhibitor pre-treated cells support the growth and survival of BRAF inhibitor sensitive cells *in vivo*

The fact that MITF heterogeneity was still maintained during the course of treatment in various experimental settings suggested

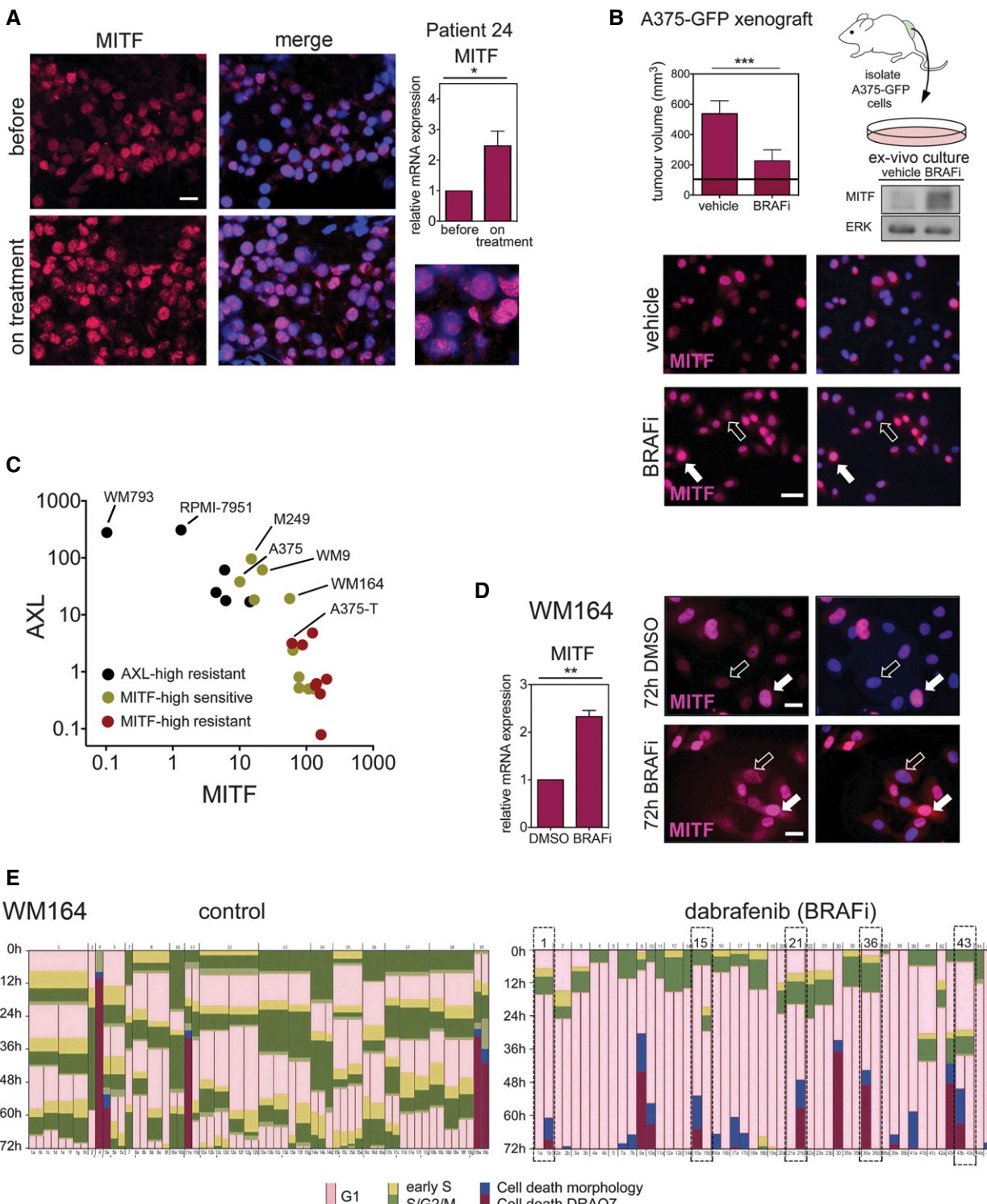

**Figure 1.**

that cells with basal MITF expression levels could withstand the drug insult in the presence of cells with up-expressed MITF levels. To test this, we "created" heterogeneous xenografts containing A375-T cells alongside A375 cells. A375-T cells display treatment-induced up-expression of MITF (Fig 2A), and as a consequence,

A375-T cells are more resistant to BRAF inhibition (Fig 2B and Smith *et al*, 2016). We injected GFP-expressing A375 cells together with RFP-expressing A375-T cells into zebrafish embryos as described (Chapman *et al*, 2014), and monitored xenograft growth. As expected, A375-T xenografts were more resistant to

Figure 1.  MITF heterogeneity is maintained during MAPK inhibitor treatment.
A    Immunofluorescence analysis for MITF (magenta) in a tumour of a patient, who had been treated with dabrafenib 2 weeks. Nuclei were stained with DAPI. Scale
     bar: 10 μm. Relative MITF mRNA expression assessed by qRT–PCR is shown. *P*: probability by paired *t*-test: *P = 0.0219. Error bars: SD of three replicate measures.
B    A375-GFP cells were isolated from xenografts grown in mice (n = 6) that had been treated with vehicle or 100 mg/kg vemurafenib (BRAFi) for 12 days (maximum
     response). The tumour volume is indicated (mean ± SD; horizontal black line, 100 mm$^3$, volume at start); *P*: probability by *t*-test: ***P = 0.001. The *ex vivo* cultures
     were analysed for MITF expression by Western blot and immunofluorescence (magenta). Nuclei were stained with DAPI. Scale bar: 20 μm (white arrows, high MITF;
     black arrows, low MITF).
C    Relative AXL and MITF expression in a panel of melanoma cell lines that have been characterized for their response to BRAF inhibition (Barretina *et al*, 2012; Garnett
     *et al*, 2012; Smith *et al*, 2013).
D    MITF immunofluorescence analysis of WM164 cells treated with DMSO or dabrafenib for 72 h. Scale bar: 10 μm. Relative MITF mRNA expression assessed by
     qRT–PCR is shown (n = 3 independent experiments; mean ± SEM). *P*: probability by paired *t*-test: **P = 0.0095 (white arrows, high MITF; black arrows, low MITF).
E    Time-lapse analysis of mKO2-hCdt1 and mAG-hGem (FUCCI) expressing WM164 melanoma cells (Haass *et al*, 2014) over 72 h. Cells were either treated with DMSO or
     dabrafenib. Dashed lines indicate cells whose daughter cells underwent different fates after exiting mitosis.

Source data are available online for this figure.

BRAF inhibition and still grew in the presence of drug (Fig 2B and C). Moreover, in heterogeneous xenografts A375-T cells could protect A375 cells from BRAF inhibitor-induced growth inhibition (Fig 2C and D), thereby maintaining heterogeneity. In addition, even in the absence of drug, the presence of A375-T cells in heterogeneous xenografts appeared to have a growth stimulating effect on A375 cells (Fig 2D). This was also seen in melanoma spheres grown in 3D dermal collagen, where A375-T had a pro-proliferative effect on A375 cells (Fig EV2).

## Paracrine protection is a general concept

To test whether the above observed co-culture protection was mediated by soluble factors being produced in response to drug treatment, we exposed A375 cells to BRAF inhibitor over the course of 14 days, and periodically collected conditioned medium at 3, 7 and 14 days. Exposing parental sensitive cells to these conditioned media reduced the efficacy of BRAF inhibition, and these effects were most significant after 7–14 days (Fig 3A). A similar effect was observed with conditioned medium from three other BRAF$^{V600E}$ melanoma cell lines WM164, M249 and WM9 (Figs 3A and EV3A). Although differing in their time course, after 14 days all cell lines displayed enhanced MITF expression and were significantly more tolerant to BRAF inhibitor than their respective parental cell lines (Fig 3A and Appendix Fig S2). The transient nature of this drug-induced secretome was seen when cells were taken off the drug, which resulted in the reduction of MITF expression and loss of the protective effect brought about by the conditioned medium (Fig EV3B and C).

To create a situation in which both cell populations are exposed to drug at the same time, but soluble factors can be effective, we co-cultured sensitive parental cells with their tolerant daughter cells separated by a permeable membrane (Fig 3B). In this setting, we could confirm that soluble factors were activating a protective signalling. Moreover, we found that pre-treated melanoma cells could produce this effect in other melanoma cell lines and that innate-resistant high MITF-expressing cell lines (Smith *et al*, 2013) can also produce such a paracrine protective effect (Fig 3C).

Thus, prolonged BRAF inhibition leads to the production of a secretome that counteracts the growth suppressive effects of MAPK pathway inhibition. While this secretome possibly acts in an autocrine fashion, we show here that it also can act in a paracrine mode, thereby protecting otherwise drug-sensitive cells.

## Paracrine protection involves re-activation of the MAPK pathway via PKC

To further dissect the mechanism of paracrine protection, we assessed MAPK pathway activation and found that the co-culture with 14 days pre-treated cells led to a partial rescue of ERK phosphorylation in sensitive cells in the presence of a BRAF inhibitor (Fig 3D). A similar effect was observed with A375 *ex vivo* cultures isolated from tumours that had regressed on BRAF inhibitor (Fig EV3D), as well as with *in vitro* generated A375-T cells (Fig EV3E). MEK inhibition could overcome the paracrine protection and ERK re-activation mediated by soluble factors (Fig EV3E). This indicated that ERK re-activation occurs upstream of MEK, and the most prominent candidate for this activation is CRAF. We thus used the pan-RAF inhibitor RAF265, which abolished the re-activation of ERK phosphorylation (Fig 3E) and completely overcame the protective effect produced by A375-T cells (Fig 3F). A similar effect was observed in other melanoma cell lines when they were treated with conditioned medium (Fig EV3F). Using specific inhibitors to identify the upstream activator of CRAF revealed that the pan-PKC inhibitor GO-6983 (PKCi) was able to overcome ERK re-activation and the protective effect produced by co-culturing A375 cells with A375-T cells (Fig 3E and F). These data strongly suggest that prolonged BRAF inhibition triggers the production of secreted factors capable of re-activating the pathway via PKC and CRAF. Indeed, treatment of melanoma cell lines with conditioned medium derived from corresponding cell lines treated for 14 days with BRAF inhibitor, resulted in an increase in the phosphorylation of proteins recognized as PKC substrates (Fig 3G).

## The secretome of BRAF inhibitor pre-treated cells is enriched in Endothelin-1

A quantitative proteomics analysis of conditioned medium from untreated A375 and A375-T cells (applying a cut-off of twofold change) identified 387 proteins that were enriched in the conditioned medium of pre-treated A375-T cells; amongst these were 77 secreted/extracellular space-signalling proteins (Fig 4A). Ingenuity Pathway Analysis identified 27 of these 77 proteins as activators of ERK (Fig 4B). Amongst these was EGF, which together with its receptor has been previously implicated in BRAF inhibitor resistance through reactivation of ERK (Girotti *et al*, 2013; Sun *et al*, 2014). Nevertheless, while EGFR expression was up-regulated in A375-T

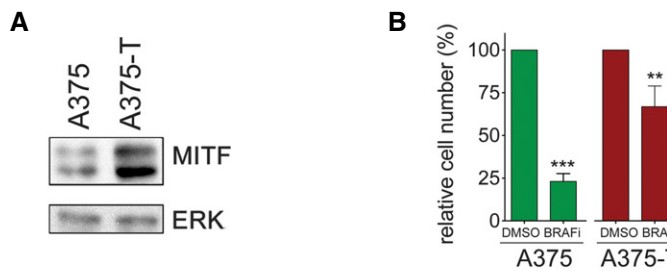

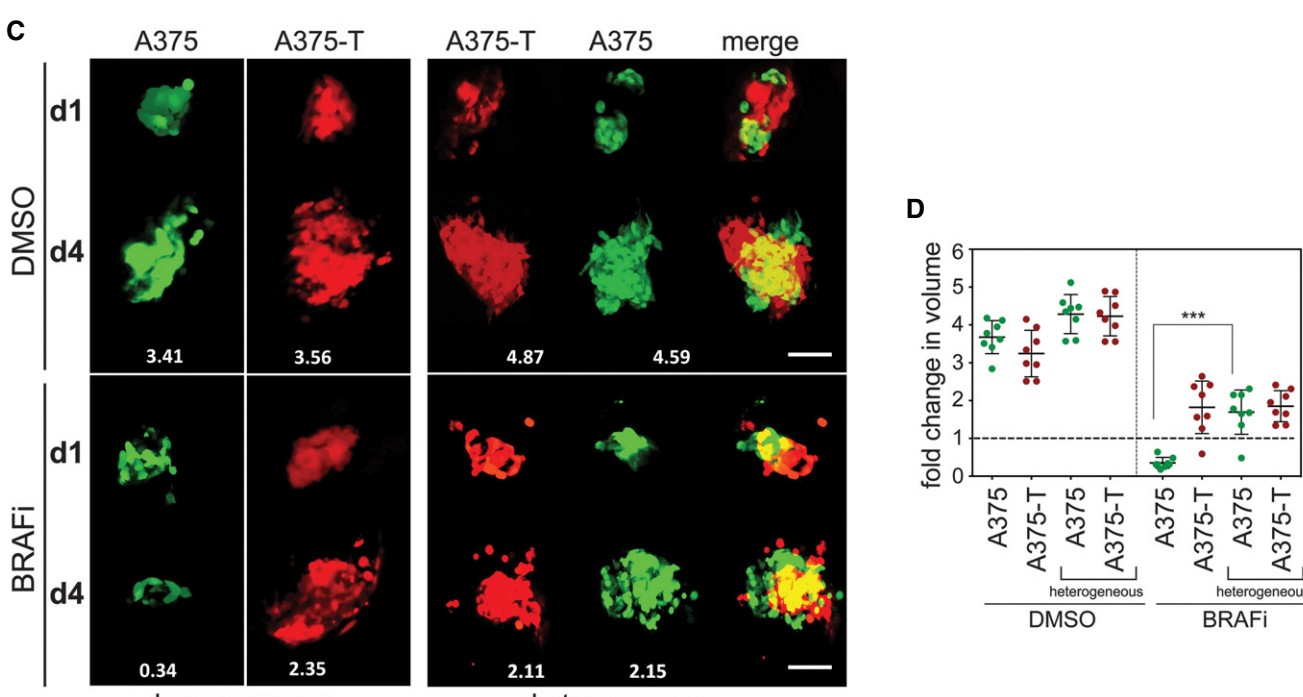

**Figure 2. MITF heterogeneity counteracts MAPK inhibitor-induced growth inhibition *in vivo*.**

A   Western blot of A375 or A375-T cells depicting basal expression of MITF. ERK2 served as loading control.

B   A375 or A375-T cells were treated with vemurafenib (BRAFi) for 72 h before relative cell number was assessed. A Western blot for MITF in A375 and A375-T cells is shown. *P*: probability by one-way ANOVA (with Tukey's *post hoc* test); ***$P < 0.0001$ (A375) and **$P = 0.0022$ (A375-T).

C   GFP-expressing A375 or RFP-expressing A375-T cells were injected into the pericardial space of zebrafish embryos, before they were treated with either vemurafenib (BRAFi) or DMSO. The total number of cells for each injection condition was 1,000 cells. Xenografts were imaged at day 1 and day 4 of drug treatment using a Leica SP5 confocal microscope, and fold change in volume of populations of GFP- or RFP-expressing cells at day 4 compared to day 1 is indicated. Scale bar: 100 μm.

D   Xenograft volumes seen in 3D images at day 1 and day 4 of treatment were quantified using Volocity® software. Fold change relative to day 1 is shown. *P*: probability by one-way ANOVA (with Tukey's *post hoc* test); ***$P < 0.0001$.

Data information: Data are pooled of $n = 3$ independent experiments and are shown as mean ± SEM.
Source data are available online for this figure.

---

cells, it was hardly detectable in A375 cells (Appendix Fig S3A), which suggests that EGF can act in an autocrine fashion on A375-T cells, but it is unlikely to contribute to the paracrine effects observed in A375 cells.

In an attempt to narrow down the list of potential ERK re-activators, we applied a predictive algorithm for the isolation of upstream regulators of the proteins enriched in the conditioned medium. This analysis identified the MiT family factors TFEB and MITF amongst the transcriptional regulators with the highest significance (Fig 4C). Indeed, MITF depletion abolished the ability of A375-T cells to enhance A375 cell survival in the presence of BRAF inhibitor

(Fig 4D). This indicated that in A375-T cells, MITF contributes to paracrine protection by regulating the abundance of relevant secreted proteins.

We therefore analysed the 27 potential ERK activators for putative MITF targets (Hoek *et al*, 2008; Strub *et al*, 2011). While several MITF regulated factors (Fig 4B, labelled red) can contribute to activation of ERK in a CRAF-dependent manner, there was no obvious link to PKC activation. We therefore extended the list of MITF targets to regulators of secreted factors. This led to the identification of endothelin-converting enzyme-1 (ECE-1, Fig 4B), whose expression in A375-T cells was indeed dependent

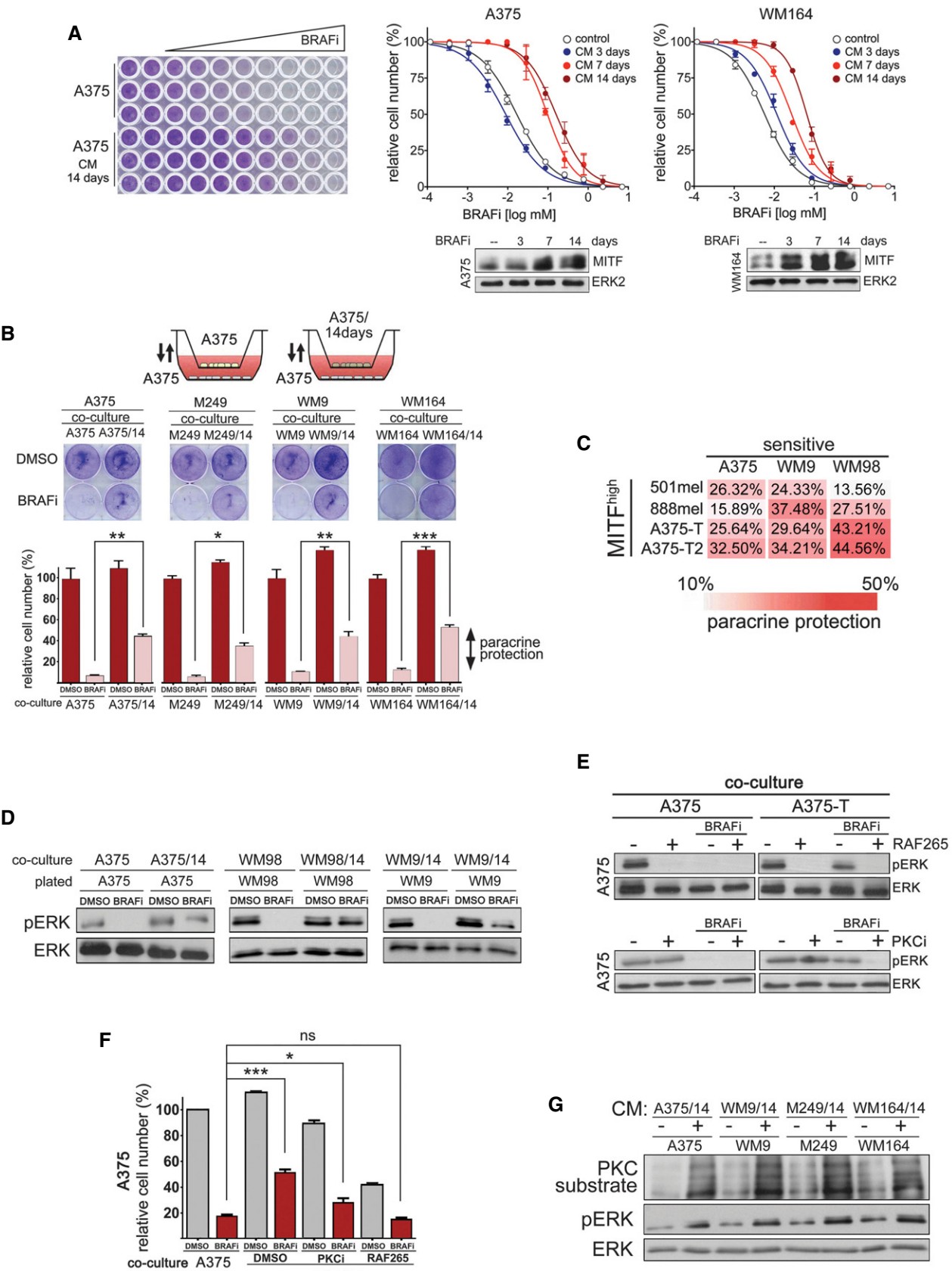

Figure 3.

**Figure 3.  Paracrine protection is a general trait of drug-tolerant melanoma cells and dependent on RAF and PKC.**

A  Dose–response curves for vemurafenib (BRAFi) in the indicated cell lines. The respective cell lines were treated with either DMEM (control) or conditioned medium, which was derived from the respective cell line treated with vemurafenib for the indicated times. Western blots for MITF with ERK2 as loading control are shown.

B  Schematic of co-culture assay of drug-tolerant and drug-sensitive melanoma cells. Drug-sensitive cells were co-cultured with either drug-sensitive cells or cells that had been pre-treated with vemurafenib for 14 days. The co-cultures were treated for 48 h with vemurafenib (BRAFi) before quantitative analysis of paracrine protection. Paracrine protection was determined as difference between co-culture with sensitive cells and drug pre-treated cells. *P*: probability by one-way ANOVA (with Tukey's *post hoc* test); **P* = 0.0022 (A375), **P* = 0.0307 (M249), **P* = 0.0084 (WM9) and ****P* < 0.0001 (WM164).

C  Analysis of paracrine protection in a panel of drug-tolerant and drug-sensitive melanoma cells. Drug-sensitive A375, WM9 or WM98 cells were co-cultured with the indicated melanoma cell cultures, and paracrine protection was determined as indicated in (B).

D  Western blot of the indicated cell cultures for pERK and total ERK. The indicated cultures were co-cultured with either untreated cells (control) or with the respective drug pre-treated cell lines. Cells were treated with DMSO or with vemurafenib (BRAFi) for 24 h.

E  Western blot of the indicated cell cultures for pERK and total ERK. Cells were either treated with DMSO or treated with vemurafenib (BRAFi) in the presence or absence of RAF265 or GO-6983 (PKCi).

F  Quantification of relative cell numbers. A375 cells were co-cultured with either A375 or with A375-T cells. DMSO-treated A375 cells in the presence of A375 cells were set at 100%. *P*: probability by *t*-test: ns *P* > 0.05, ****P* < 0.0001 and **P* = 0.0312.

G  Western blot for pERK and proteins that represent PKC substrates. ERK served as loading control. The indicated cell lines had been either left untreated or were treated with conditioned medium derived from the respective drug pre-treated (14 days) cell lines.

Data information: Data are pooled of *n* = 3 independent experiments and are shown as mean ± SEM.
Source data are available online for this figure.

---

on MITF (Fig 4E) and was up-regulated in MAPK inhibitor-treated cells (Appendix Fig S3B). What made ECE1 such an attractive regulator is that it produces biologically active endothelin-1 (EDN1), a crucial factor for the development of the melanocytic lineage (Imokawa *et al*, 2000; Saldana-Caboverde & Kos, 2010) that activates MAPK signalling through EDNRB in a PKC/CRAF-dependent manner (see Fig 4B). Indeed, when we analysed the conditioned medium of A375-T cells using a specific ELISA, we could detect considerable levels of EDN1, which were profoundly reduced when ECE1 activity was inhibited (Fig 4F). Thus, EDN1 is produced by A375-T cells and the most likely reason for not being able to detect EDN1 in the mass spectrometry analysis could be the molecular weight cut-off (3–5 kD) we had chosen, as the mature EDN1 peptide consists of only 21 aa. Nevertheless, an involvement of EDN1 in drug-induced paracrine tolerance was further supported by the fact that its levels were also increased in the medium of all cell lines that had been treated with BRAF inhibitor for 14 days (Fig 4G).

**Endothelin-1 antagonizes BRAF inhibition via PKC activation**

While the increase in EDN1 protein in the conditioned medium could be solely due to the enhanced processing by ECE1, we wondered whether EDN1 expression itself was also up-regulated in response to long-term treatment with MAPK inhibitors. Indeed, the analysis of two A375-T cell cultures for EDN1 protein revealed enhanced expression compared to A375 cells (Fig 5A). This increase was also seen at mRNA level in A375-T cultures as well as in a whole range of BRAF inhibitor-treated melanoma cells including *ex vivo* cultures (Appendix Fig S4A–C). Moreover, MITF depletion in A375-T cells resulted in a significant reduction in EDN1 mRNA and protein expression (Fig 5B), and overexpression of MITF resulted in increased EDN1 expression (Appendix Fig S4D). This indicated that MITF is not only involved in the production of the mature peptide via ECE1, but also in the regulation of EDN1 expression.

Exposure of A375 cells to recombinant active EDN1 led to ERK activation in a dose-dependent manner (Appendix Fig S4E). As

---

**Figure 4.  MITF produces paracrine protection through a secretome containing EDN1.**

A  Schematic presentation of groups of proteins detected by quantitative mass spectrometry to be enriched in conditioned medium of A375-T cells when compared to A375 cells.

B  Ingenuity Pathway Analysis of factors in the extracellular space for involvement in the activation of ERK. MITF targets are indicated in red font.

C  Ingenuity Upstream Regulator Analysis shows the potential transcriptional regulators that can explain changes observed in secreted proteins. Proteins that were up-regulated in the media of drug-treated cells (FC > 2, FDR < 0.05) were selected for IPA analysis, and upstream transcriptional driver were identified. The *Z*-score indicates activation states of predicted regulators with positive values corresponding to activated transcription regulator activating gene expression.

D  Quantification of relative cell number and analysis of ERK phosphorylation of A375 cells when co-cultured with either A375 or with A375-T cells. DMSO-treated A375 cells in the presence of A375 cells = 100%. Before co-culture, A375-T cells were treated with a control or two different MITF-specific siRNAs (siMI1, siMI3). A Western blot demonstrating the degree of MITF knockdown is shown. ERK served as loading control. *P*: probability by one-way ANOVA (with Tukey's *post hoc* test); ns *P* > 0.05, **P* = 0.0047 (A375 co-culture vs. A375-T co-culture).

E  qRT–PCR for ECE1 expression in the indicated melanoma cell lines treated with a control (sicon) or two different MITF-specific siRNAs (siMI1, siMI3). A Western blot demonstrating the degree of MITF knockdown is shown. ERK served as loading control. *P*: probability by one-way ANOVA (with Tukey's *post hoc* test); ****P* < 0.0001 (A375-T siMi1 and siMi3, A375-T2 siMi1), ****P* = 0.0003 (A375T2-siMI3).

F  A375-T cells were left untreated or treated for 24 h with 15 μM ECE1 inhibitor CGS 35066 before EDN1 levels in the medium were analysed by ELISA. *P*: probability by *t*-test: ****P* < 0.0001.

G  ELISA measuring EDN1 concentrations in the medium of the indicated cell lines. Cells were either untreated or had been treated with BRAFi for 14 days. *P*: probability by one-way ANOVA (with Sidak's *post hoc* test); ****P* < 0.0001 (WM9, M249), **P* = 0.0013 (WM98) and **P* = 0.0238 (A375).

Data information: Data are pooled of *n* = 3 independent experiments and are shown as mean ± SEM.
Source data are available online for this figure.

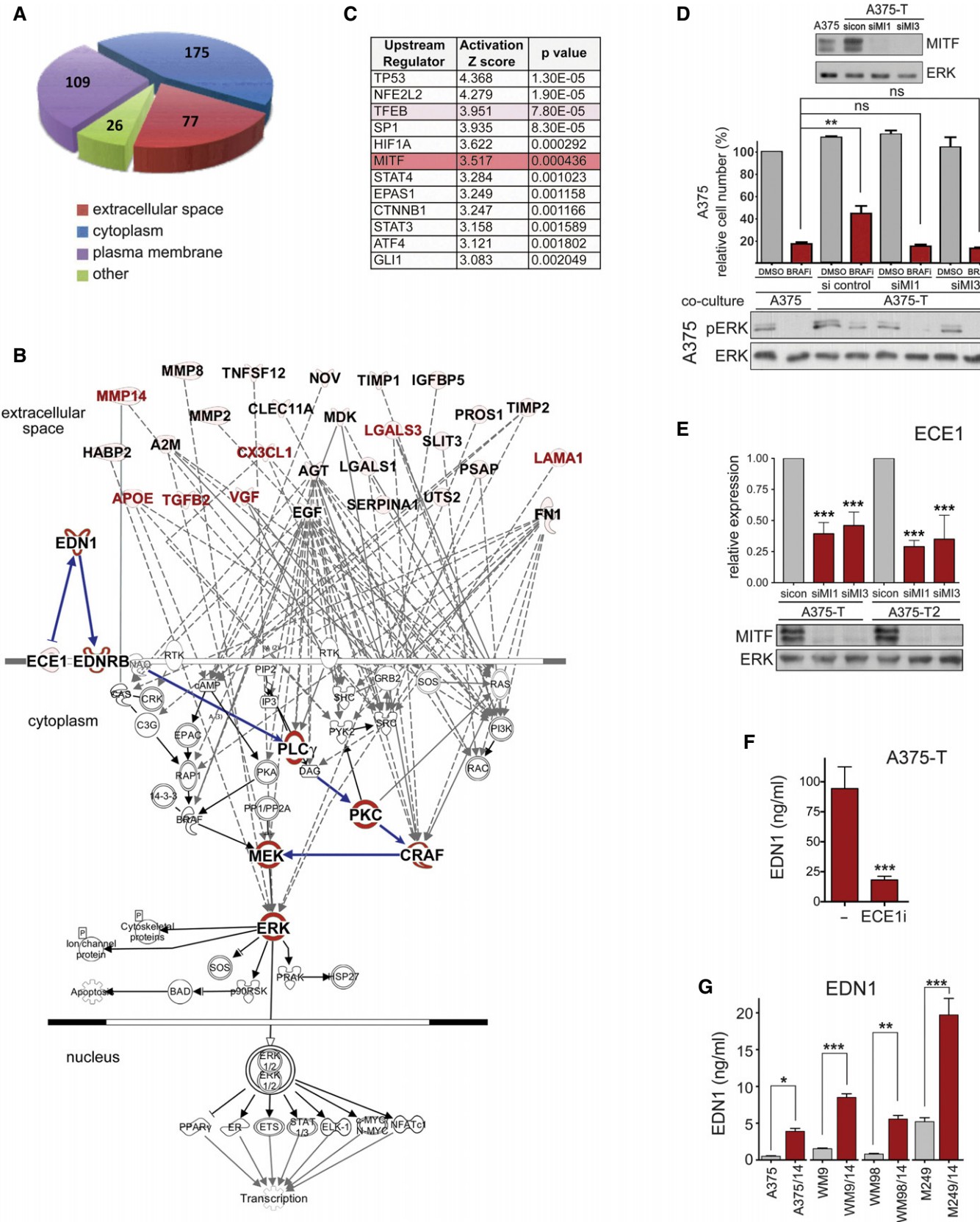

Figure 4.

**Figure 5.    MITF produces paracrine protection through EDN1-mediated PKC activation.**

A    Western blot for EDN1 and MITF expression in A375, A375-T and A375-T2 cells. Beta-tubulin and ERK2 served as loading control, respectively.
B    qRT–PCR for EDN1 expression and Western blot for EDN1 in A375-T cells after treatment with either a control siRNA (sicon) or two different MITF-specific siRNAs (siMI1, siMI3). *P*: probability by one-way ANOVA (with Tukey's *post hoc* test); ***$P < 0.0001$.
C    A375 cells were stimulated with EDN1 for 8 h and analysed for phosphorylation of PKC substrates, pERK and total ERK on a Western blot. Cells were treated with DMSO or GO-6983 (PKCi) before analysis (upper panel), or vemurafenib (BRAFi, middle panel) or RAF265 (lower panel).
D    Quantification of relative cell number of A375 cells when treated with EDN1 in the absence or presence of vemurafenib (BRAFi) and GO-6983 (PKCi) or RAF265. *P*: probability by one-way ANOVA (with Tukey's *post hoc* test); ns $P > 0.05$, ***$P = 0.0006$.
E    Quantification of relative cell number of A375-pLKO or A375-shEDNRB cells when treated with EDN1 in the absence or presence of BRAFi. *P*: probability by one-way ANOVA (with Tukey's *post hoc* test); ns $P > 0.05$, ***$P = 0.0003$.
F    GFP-A375 pLKO, GFP-A375 shEDNRB or RFP-A375-T cells were injected into the pericardial space of zebrafish embryos, and the embryos were treated with either vemurafenib (BRAFi) or the vehicle DMSO. Images at day 1 and day 4 of treatment are shown, and fold change in volume at day 4 compared to day 1 was quantified. Scale bar: 100 μm. *P*: probability by one-way ANOVA (with Tukey's *post hoc* test); ns $P > 0.05$, *$P = 0.026$ (DMSO), ***$P < 0.0001$ (BRAFi).
G    Quantification of relative cell number of A375 cells when co-cultured with either A375 or with A375-T cells in the absence or presence of BRAFi, alone or in combination with bosentan or macitentan. *P*: probability by one-way ANOVA (with Tukey's *post hoc* test); ***$P < 0.0001$ (all).

Data information: Data are pooled of $n = 3$ independent experiments and are shown as mean ± SEM.
Source data are available online for this figure.

expected, EDN1 increased the phosphorylation of PKC substrates in addition to ERK, which was however completely abolished in the presence of a PKC inhibitor (Fig 5C). The EDN1-mediated ERK activation was more obvious in the presence of a BRAF inhibitor, where it could maintain phospho-ERK levels in a RAF-dependent manner (Fig 5C). Accordingly, EDN1 protected against inhibitor-induced growth inhibition, and in line with a role for PKC, RAF and MEK, this protection was reduced in the presence of respective inhibitors (Figs 5D and EV4A). Furthermore, RNAi-mediated EDN1 depletion from A375-T cells used in transwell co-culture assays significantly reduced paracrine protection (Fig EV4B), and similar effects were seen with an EDN1 neutralizing antibody (Fig EV4C).

**EDNRB is required for paracrine protection**

EDN1 signals through endothelin receptors, of which EDNRB is essential for melanocyte development (Saldana-Caboverde & Kos, 2010), and its expression is highly enriched in melanoma cells (Fig EV4D). Indeed, EDNRB depletion through RNAi prevented EDN1 from protecting melanoma cells from BRAF inhibition (Fig 5E). To assess the relevance of EDNRB for paracrine protection *in vivo*, we co-injected A375-T cells with A375 control cells or with A375-shEDNRB cells into zebrafish embryos, and monitored cell death within tumours in the absence or presence of BRAF inhibitor. As seen for A375 cells, BRAF inhibition reduced the volume of A375-shEDNRB xenografts (Fig 5F). However, in contrast to what we observed in A375 cells, A375-T cells were not able to protect A375-shEDNRB cells from BRAF inhibition (Fig 5F), indicating that EDNRB-mediated signalling plays an important role in paracrine protection to MAPK pathway inhibition in melanoma cells.

To assess the pharmacological intervention of this signalling, we used EDN receptor (EDNR) antagonists bosentan and macitentan, which are currently used to treat pulmonary artery hypertension, but are also being trialled in solid cancers (Rosano *et al*, 2013). Both drugs could overcome the paracrine protection brought about by A375-T cells (Fig 5G). Bosentan also partially restored the sensitivity of a panel of parental cell lines to BRAF inhibition when they were "protected" by conditioned medium from pre-treated cells (Fig EV4E). Finally, macitentan overcame paracrine protection and ERK re-activation induced by conditioned medium from A375 cells ectopically overexpressing MITF (Appendix Fig S4F and G), further

supporting the link between MITF, EDNR signalling and paracrine protection.

**Targeting EDNR enhances BRAF inhibitor efficacy *in vivo* and suppresses enrichment for AXL-high cell populations**

Our data suggest that blocking EDNRB-mediated signalling could improve BRAF inhibitor responses by preventing EDN1-mediated paracrine signalling. In order to test whether therapeutic intervention with EDNRB signalling represents a relevant strategy for melanoma patients, we analysed melanoma samples isolated from patients when they had been on treatment for 2 weeks. In line with our previous findings that MITF is up-regulated on treatment (Smith *et al*, 2016), we found that EDN1 as well as EDNRB expression increased in melanomas of patients ($n = 22$) on treatment (Fig 6A). Moreover, EDN1 is up-regulated in short-term cultures from melanomas from patients on treatment (see Appendix Table S2 for patient details), and these cultures also provide paracrine protection (Fig EV4F and G), supporting the relevance of our findings in patients.

To assess the efficacy of targeting EDNRB signalling, we treated mice bearing A375 tumours either with BRAF inhibitor alone or a BRAF inhibitor/bosentan combination and found that growth was significantly reduced with the combination treatment (Fig 6B). Under the chosen conditions, ERK phosphorylation and activity (assessed through the surrogate marker DUSP6) was reduced with BRAF inhibitor (Fig 6C), but the combination treatment led to a significant further reduction, supporting the idea that EDNRB signalling contributes to ERK activation within these tumours. The reduced ERK activity in BRAF inhibitor-treated tumours correlated with increased MITF and EDN1 expression (Fig 6D). However, both EDN1 and MITF up-regulation was further enhanced with the BRAF inhibitor/bosentan combination treatment (Fig 6D). While this is in line with the observed strong reduction in ERK activation, it demonstrates that, although EDN1 is up-regulated, antagonizing EDNR signalling can severely block tumour growth.

Because on BRAF inhibitor monotherapy tumour growth had resumed despite ERK being still inhibited, we wanted to see whether these MITF-high tumours are enriched for AXL-high cell populations. Indeed, AXL expression was increased in tumours growing on BRAF inhibitor monotherapy (Fig 6E). Intriguingly however,

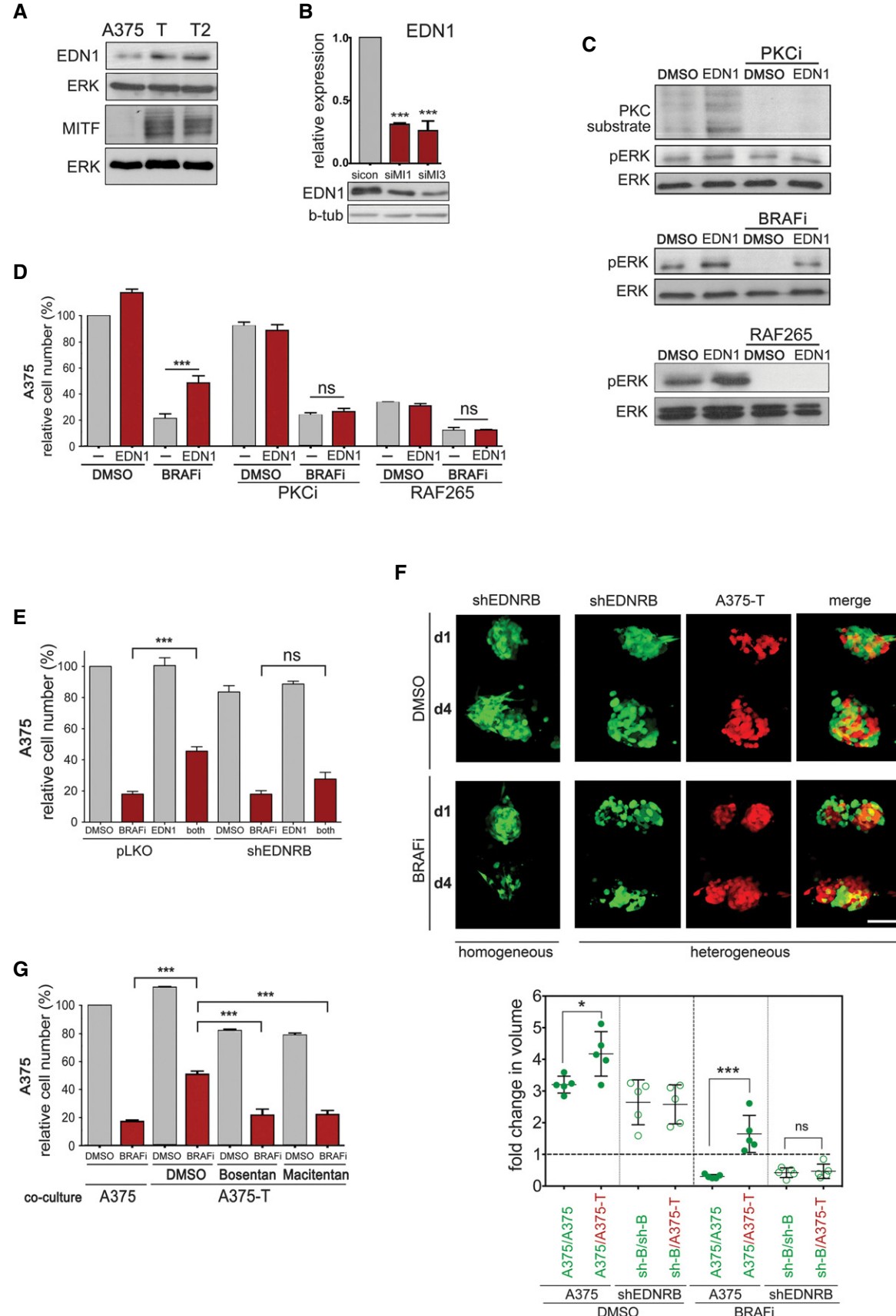

**Figure 5.**

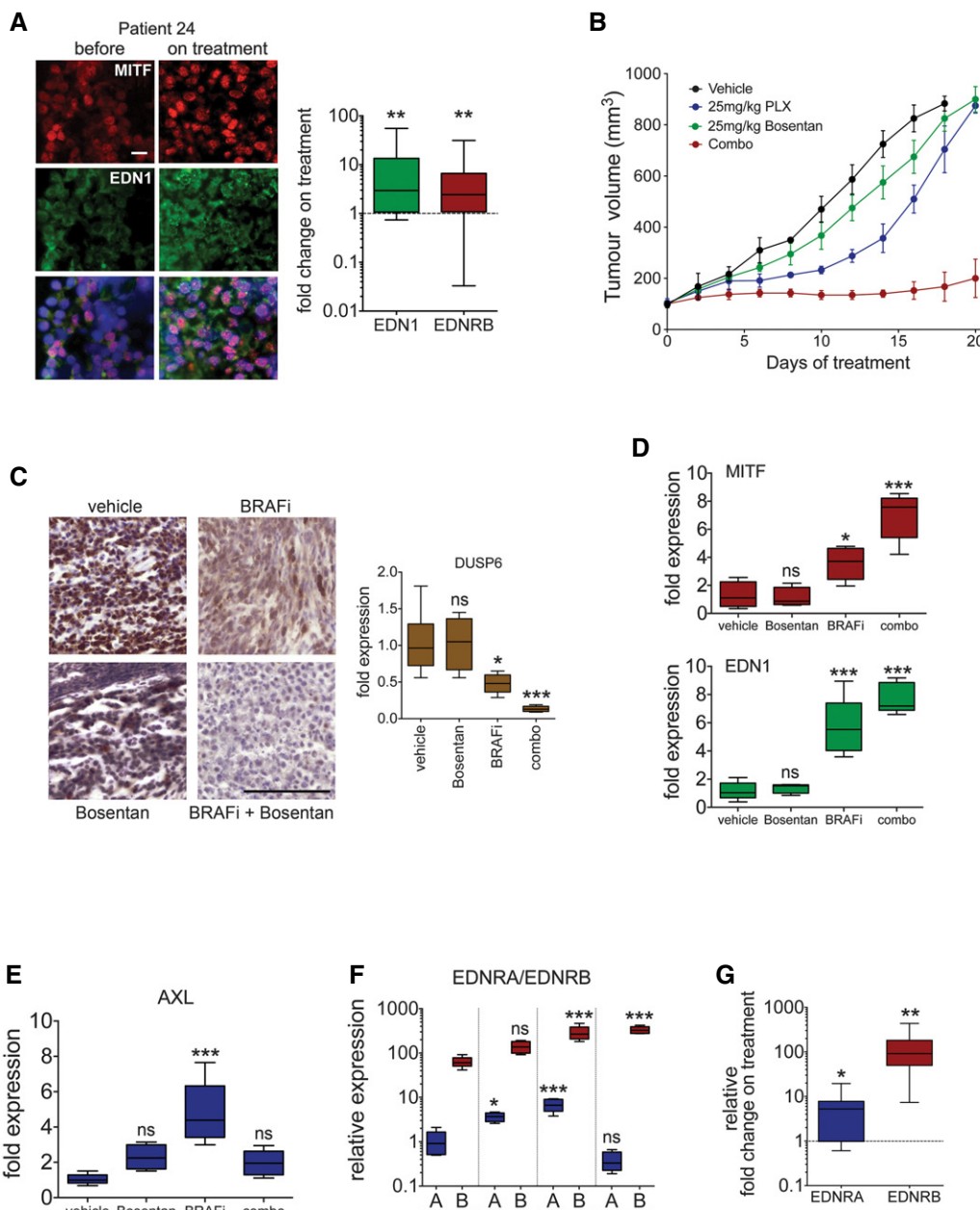

**Figure 6.   Inhibition of EDNR signalling *in vivo* reduces tumour growth and counteracts AXL up-regulation.**

A   Analysis of EDN1 and EDNRB expression in patients. Immunofluorescence analysis for MITF and EDN1 expression in the tumour of patient 24 before and on treatment. Scale bar: 10 μm. EDN1 and EDNRB qRT–PCR analysis in tumours of patients on treatment with either vemurafenib alone or a dabrafenib/trametinib combination (*n* = 22). *P*: probability by *t*-test; \*\**P* = 0.0064 (EDN1) and \*\**P* = 0.0033 (EDNRB).

B   Nude mice bearing A375 tumours were treated (*n* = 5–6 mice per group) with vehicle, vemurafenib (25 mg/kg/qd) or bosentan (25 mg/kg/qd) alone or in combination for 20 days. Data are presented as mean tumour volumes ± SEM.

C   Phospho-ERK IHC and qRT–PCR for DUSP6 from tumours corresponding to the experiment described in (B). Scale bar: 200 μm. *P*: probability by one-way ANOVA (with Tukey's *post hoc* test); ns *P* > 0.05, \*P = 0.0186 (BRAFi) and \*\*\*P = 0.0004 (combo).

D   qRT–PCR for MITF and EDN1 from tumours corresponding to the experiment described in (B). *P*: probability by one-way ANOVA (with Tukey's *post hoc* test); ns *P* > 0.05, \*P = 0.0179 (MITF-BRAFi), \*\*\*P < 0.0001 (MITF-combo), \*\*\*P < 0.0001 (EDN1-BRAFi) and \*\*\*P < 0.0001 (EDN1-combo).

E, F   qRT–PCR for AXL, and EDNRA and EDNRB from tumours corresponding to the experiment described in (B). *P*: probability by one-way ANOVA (with Tukey's *post hoc* test); ns *P* > 0.05, \*P = 0.0341 (EDNRA-Bosentan) and \*\*\*P < 0.0001 (AXL-BRAFi, EDNRA-BRAFi, EDNRB-BRAFi and EDNRB-combo).

G   qRT–PCR for EDNRA and EDNRB in tumours of patients on treatment with either vemurafenib alone or a dabrafenib/trametinib combination. Relative basal expression of EDNRA and EDNRB was considered. *P*: probability by *t*-test; \*P = 0.0116 (EDNRA) and \*\*P = 0.0039 (EDNRB).

Data information: Data are pooled of *n* = 3 independent experiments and are shown as mean ± SEM. Box and whiskers plots with median (horizontal line), second and third quartiles (box limits) and min and max values (error bars).

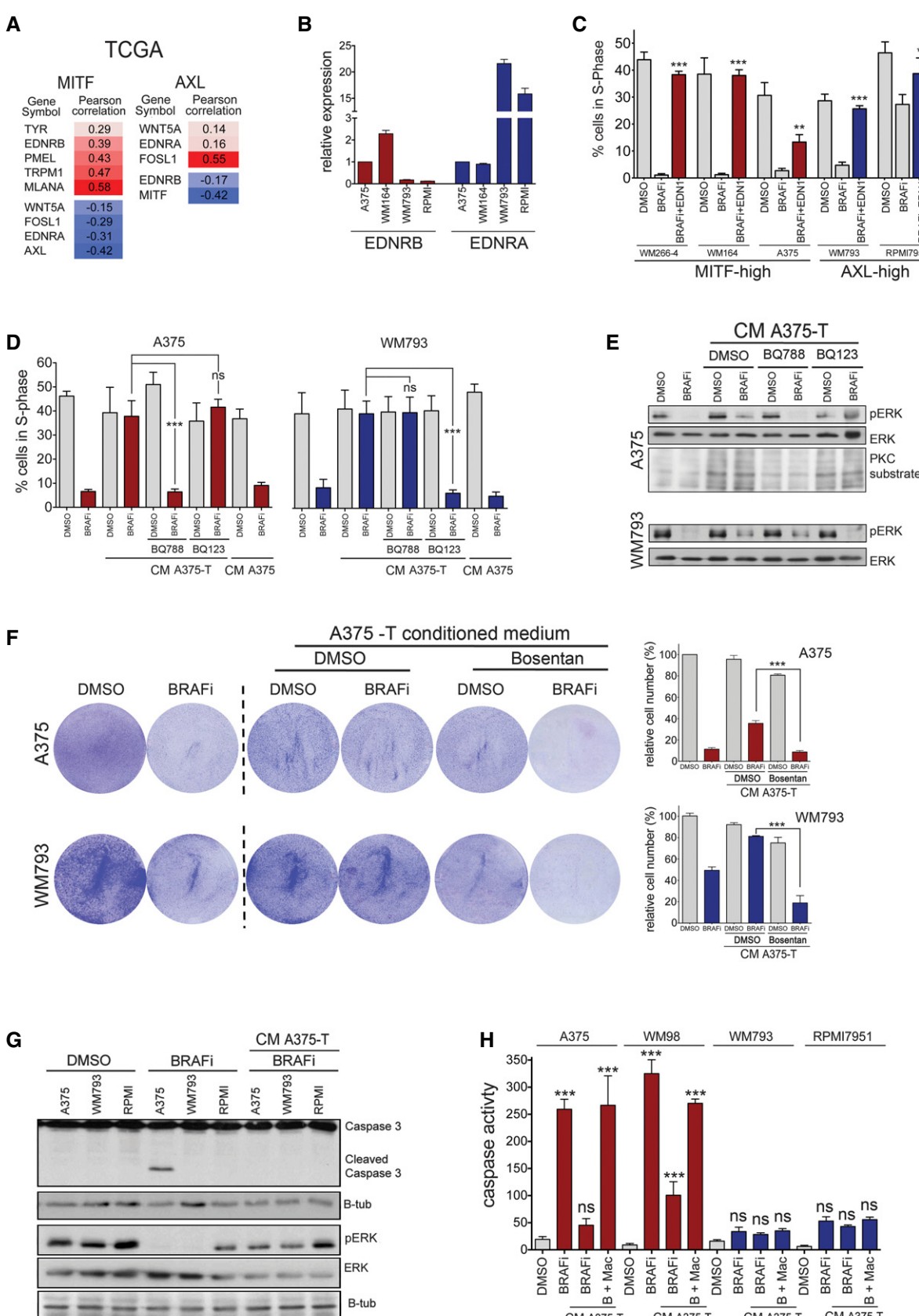

Figure 7.

◀

**Figure 7.  EDNR signalling is required for proliferation in AXL-high cells.**

A    Correlation analysis for the expression of the indicated genes using the TCGA-melanoma dataset (TCGANetwork, 2015).

B    Real-time qPCR of EDNRA and EDNRB in the indicated MITF-high and AXL-high cell lines.

C    Analysis of cells in S-phase. The indicated cell lines were treated with vemurafenib and 20 ng/ml EDN1 either alone or in combination for 24 h, and 4 h before analysis EdU was added to the cultures. *P*: probability by one-way ANOVA (with Tukey's *post hoc* test); ***$P < 0.0001$ (WM266-4, WM164, WM793), **$P = 0.009$ (A375), *$P = 0.0445$ (RPMI7951).

D    Analysis of cells in S-phase. The indicated cell lines were treated with vemurafenib (BRAFi), BQ788 or BQ123 either alone or in combination and in the presence of conditioned medium from A375-T cells for 24 h, and 4 h before analysis EdU was added to the cultures. The conditioned medium from A375 cells was used as control. *P*: probability by *t*-test; ns $P > 0.05$, ***$P = 0.0007$ (A375, BRAFi/BQ788) and ***$P = 0.0009$ (WM793, BRAFi/BQ123).

E    Western blot of A375 and WM793 cells in the absence or presence of conditioned medium from A375-T cells treated with DMSO or BRAF inhibitor for phospho-ERK and PKC substrates as indicated. ERK2 served as loading control.

F    Quantification of relative cell numbers. A375 or WM793 cells were either left untreated or were treated with 0.5 μM vemurafenib (BRAFi) and bosentan either alone or in combination in the presence of conditioned medium from A375-T cells for one week. DMSO-treated A375 cells were set at 100%. *P*: probability by one-way ANOVA (with Tukey's *post hoc* test); ***$P < 0.0001$.

G    Western blot for the indicated proteins of the indicated cell lines with DMSO, vemurafenib (BRAFi) or BRAFi in the presence of conditioned medium from A375-T cells.

H    Analysis for apoptosis using an incucyte® caspase-3/7 apoptosis assay reagent. The indicated cell lines were treated with DMSO, vemurafenib (BRAFi) alone or with BRAFi +/− macitentan (Mac) in the presence of conditioned medium from A375-T cells and apoptosis activity was measured over time; end-point values at 48 h are shown. *P*: probability by one-way ANOVA (with Tukey's *post hoc* test); ns $P > 0.05$, ***$P < 0.0001$.

Data information: Data are pooled of $n = 3$ independent experiments and are shown as mean ± SEM.
Source data are available online for this figure.

addition of bosentan overcame this increase in AXL expression, which suggested that blocking EDNR signalling also affects AXL-high cells.

This was surprising as EDNRB expression is linked to the MITF-high signature and not found in the AXL-high signature (Hoek *et al*, 2006). However, it has to be noted that EDN1 not only activates EDNRB but is also a strong activator of EDNRA. We therefore assessed EDNRA expression in the A375 xenografts and found that although it was expressed at much lower levels than EDNRB, its expression was up-regulated in BRAF inhibitor-treated tumours (Fig 6F). A similar situation was observed in melanomas from patients, where EDNRA expression levels were also lower than EDNRB levels and were increased on treatment (Fig 6G). This suggested that EDNRA could also contribute to EDN1 signalling in patients during therapy. Intriguingly, in xenografts, the BRAF inhibitor/bosentan combination treatment led to a significant suppression of EDNRA expression within the residual tumours (Fig 6F), which correlates with the drop in AXL expression (Fig 6E).

## AXL-high cell populations require EDN1 to overcome BRAF inhibition

As mentioned above EDNRB but not EDNRA expression is linked to the MITF-high signature. Because we saw a correlation of AXL expression with EDNRA expression in the xenografts, we interrogated publicly available gene expression data, including the TCGA-melanoma as well as two melanoma cell line datasets (Barretina *et al*, 2012; Garnett *et al*, 2012; TCGANetwork, 2015). We found that in the TCGA dataset MITF was positively correlated with EDNRB, but negatively correlated with EDNRA expression (Fig 7A). On the other hand, EDNRA expression positively correlates with AXL, and EDNRA and EDNRB expression are inversely correlated (Figs 7A and EV5A and B). We could further confirm the mutual exclusion of MITF/EDNRB and AXL/EDNRA expression in a panel of melanoma cell lines (Fig 7B).

The specific expression of EDNRA in AXL-high cells and the effect of the BRAF inhibitor/bosentan combination therapy on these cells suggested that EDNRA signalling is relevant for the growth of

AXL-high cells. Nevertheless, in contrast to MITF-high cells, recombinant EDN1 alone did not increase basal proliferation of these cells (not shown). However, in the presence of BRAF inhibitor, which produced a significant reduction in cells in S-phase, EDN1 provided a clear advantage and stimulated cell cycle progression (Fig 7C). A similar protection was seen in AXL-high cells with EDN1-containing conditioned medium from A375-T cells, whereby the EDNR antagonist bosentan overcame the protective effect of the medium (Fig EV5C). It should be mentioned that while bosentan is a pan EDNR antagonist, its affinity for EDNRA is ~20-fold higher than for EDNRB (Clozel *et al*, 1994), explaining why it also affects AXL-high cells. However, to further dissect the specific involvement of the different receptors in EDN1 signalling, we used inhibitors specifically antagonizing EDNRA (BQ123) or EDNRB (BQ788). This revealed that EDNRB was required in MITF-high cells and EDNRA was required in AXL-high cells to transmit the paracrine protection brought about by conditioned medium (CM) from A375-T cells (Fig 7D and Appendix Fig S5). The paracrine effect was linked to ERK phosphorylation, which was induced by the CM from A375-T cells, whereby BQ788 overcame this activation in MITF-high A375 cells in BQ123 in AXL-high WM793 cells, respectively (Fig 7E).

In line with high AXL expression, WM793 cells were more resistant to long-term BRAF inhibitor treatment when compared to MITF-high A375 cells (Fig 7F). However, the inhibitor still reduced the cell number over time (Fig 7F), which is entirely in line with observations in other AXL-high cell lines such as 1205Lu, IGR39, 294T and A2058 (Tsai *et al*, 2008; Beaumont *et al*, 2016; Tirosh *et al*, 2016). It appears that AXL-high cells are less responsive to BRAF inhibition, because despite slowing down cell cycle progression and reduce ERK activity (see Fig 7C and E), BRAF inhibition does not induce apoptosis in AXL-high cells (Fig 7G and H). Nevertheless, similar to what is seen in MITF-high cells, conditioned medium from BRAF inhibitor pre-treated melanoma cells profoundly protects AXL-high cells from the growth inhibitory effect of BRAF inhibition (Fig 7F). However, bosentan overcomes this paracrine protection in AXL-high cells, resulting in significant reduction in cell number and this is related to reduced ERK activity (Figs 7F and EV5D). Again use of BQ123, BQ788 as well as a specific

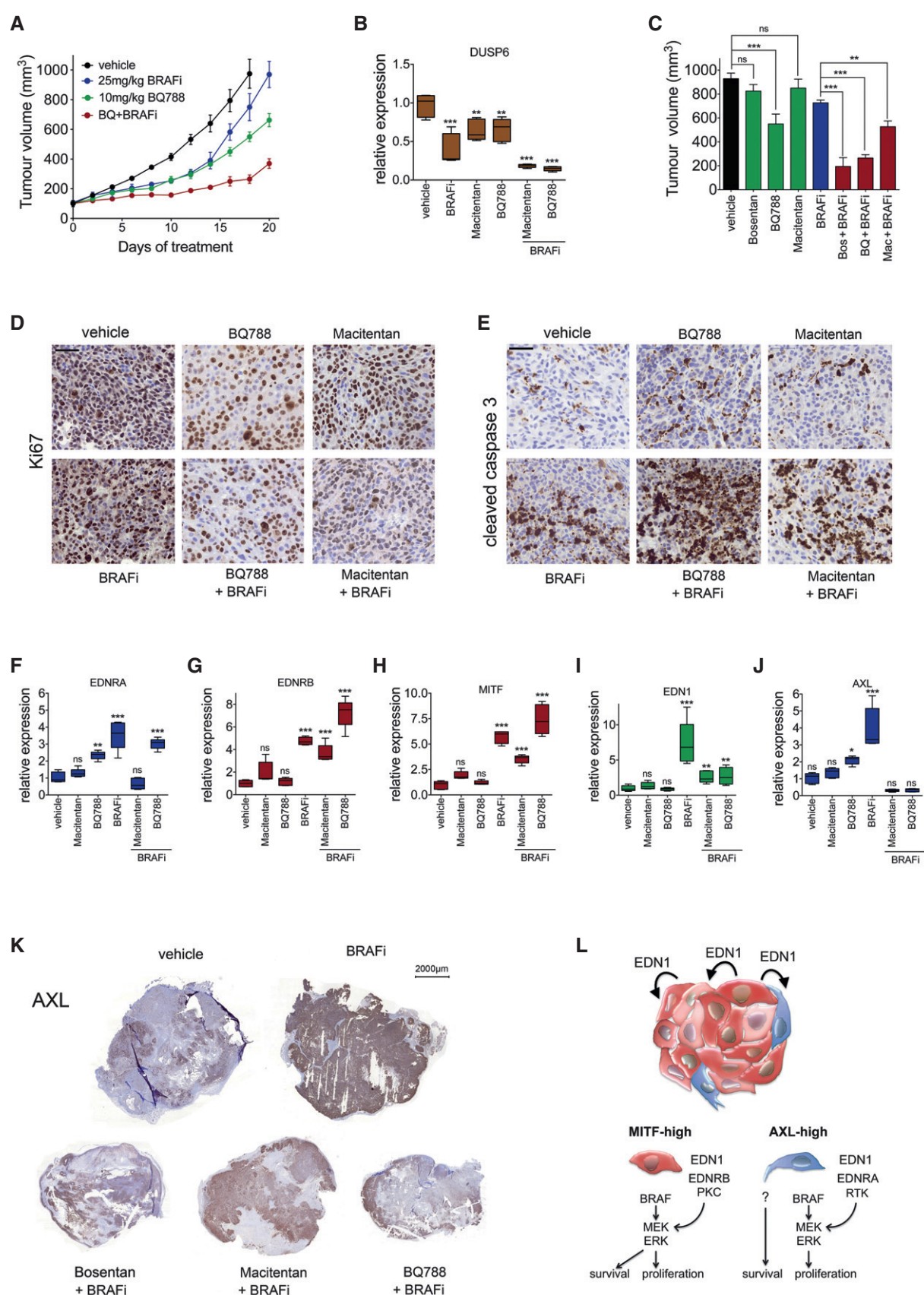

**Figure 8.**

**Figure 8.   EDNR signalling is required for proliferation in AXL-high cells.**

A   Nude mice bearing A375 tumours were treated (*n* = 5–6 mice per group) with vehicle, vemurafenib (25 mg/kg/qd) or BQ788 (10 mg/kg/qd) alone or in combination for 20 days.

B   qRT–PCR for DUSP6 from tumours corresponding to the experiment described in (A) and Fig EV6A. *P*: probability by one-way ANOVA (with Tukey's *post hoc* test); ***$P$ < 0.0001 (BRAFi, BRAFi + Macitentan, BRAFi + BQ788), **$P$ = 0.068 (Macitentan) and **$P$ = 0.082 (BQ788). Error bars, min and max values; box limits, second and third quartiles; horizontal line, median.

C   A375 tumour volume on day 18 of treatment with the indicated regimes. *P*: probability by one-way ANOVA (with Tukey's *post hoc* test); ns $P$ > 0.05, ***$P$ = 0.0004 (BQ788), ***$P$ < 0.0001 (Bos + BRAFi, BQ + BRAFi), **$P$ = 0.009 (Mac + BRAFi).

D   IHC for Ki67 in A375 xenografts from mice treated as indicated. Scale bar: 50 μm.

E   IHC for cleaved caspase-3 in A375 xenografts from mice treated as indicated. Scale bar: 50 μm.

F–J   qRT–PCR for EDNRA, EDNRB, MITF, EDN1 and AXL expression in tumours corresponding to the experiment described in (A) and Fig EV6A. *P*: probability by one-way ANOVA (with Tukey's *post hoc* test); (F) EDNRA: ns $P$ > 0.05, **$P$ = 0.0010 (BQ788), ***$P$ < 0.0001 (BRAFi) and ***$P$ < 0.0001 (BQ788 + BRAFi). (G) EDNRB: ns $P$ > 0.05, ***$P$ < 0.0001 (BRAFi), ***$P$ = 0.0002 (Mac + BRAFi) and ***$P$ < 0.0001 (BQ788 + BRAFi). (H) MITF: ns $P$ > 0.05, ***$P$ < 0.0001 (BRAFi), ***$P$ = 0.0001 (Mac + BRAFi), and ***$P$ < 0.0001 (BQ788 + BRAFi). (I) EDN1: ns $P$ > 0.05, ***$P$ < 0.0001 (BRAFi), **$P$ = 0.0331 (Mac + BRAFi) and **$P$ = 0.0225 (BQ788 + BRAFi). (J) AXL: ns $P$ > 0.05, *$P$ = 0.0462 (BQ788) and ***$P$ < 0.0001 (BRAFi).

K   IHC for AXL of A375 xenografts from mice treated as indicated. Scale bar: 2,000 μm.

L   Model of EDN1-mediated paracrine protection. EDN1 induces re-activation ERK in the presence of BRAF inhibitor. In MITF-high cells, ERK regulates proliferation and survival, but in AXL-high cells ERK only stimulate proliferation.

Data information: Data are presented as mean ± SEM.

EDN1-blocking antibody confirmed the contribution of the individual EDNRs to paracrine protection (Fig EV5E–G).

In order to dissect how EDNRA signals to ERK and hence regulates cell growth, we used different kinase inhibitors and found that while a PKC inhibitor did not impact on the protective function of EDN1, inhibiting RAF kinases with RAF265 or blocking RTK activity using the pan RTK inhibitor dovitinib overcame this protection (Fig EV5H and I). This suggests a crosstalk between EDNRA and RTK signalling as it has been described previously (Harada *et al*, 2014; Harun-Or-Rashid *et al*, 2016; Moody *et al*, 2017), which ultimately activates ERK through RAF.

**Targeting EDNRA or EDNRB reduces AXL-high populations *in vivo***

To further assess the specific contribution of EDNRB and EDNRA signalling to the response to BRAF inhibition *in vivo*, we treated mice bearing A375 tumours either with BQ788, which has a 1,000-fold higher affinity for EDNRB than EDNRA (Okada & Nishikibe, 2002) or with macitentan with an approximately 800-fold higher affinity for EDNRA than EDNRB (Boss *et al*, 2016). The EDNR inhibitors were applied either alone or in combination with BRAF inhibitor (Figs 8A and EV6A). After 20 days, tumours treated with BRAF inhibitor alone had resumed growth, whereas ERK activity was still moderately inhibited as seen by the reduced expression of DUSP6 (Fig 8B). The combination of BQ788 with BRAF inhibitor led to a significant reduction in tumour growth, which correlated with strong suppression of DUSP6 expression (Fig 8A–C). The effect of macitentan in the combination with BRAF inhibitor was weaker, but it still led to a significantly stronger reduction in tumour growth compared to BRAF inhibitor monotherapy (Figs 8C and EV6A). Overall, when compared to BRAF inhibitor alone, the combination with each of the three EDNR inhibitors bosentan, macitentan and BQ788 reduced proliferation, measured through Ki67 staining (Figs 8D and EV6B), and increased cell death, assessed by cleaved caspase-3 staining (Figs 8E and EV6C).

The addition of macitentan to the BRAF inhibitor reduced the expression of EDNRA within the tumours (Fig 8F), suggesting that through its affinity for EDNRA macitentan targets the AXL/EDNRA-expressing cells. Addition of BQ788 only slightly reduced EDNRA expression, which was still elevated (Fig 8F). This observation

would be in line with the specific activity of BQ788 towards EDNRB-expressing but not towards EDNRA-expressing cells. Similar to what we observed with bosentan, BRAF inhibitor/BQ788-treated tumours displayed increased expression of MITF and EDNRB (Fig 8G and H). This response might be due to the fact that ERK is effectively inhibited (Fig 8B), leading to MITF up-expression and eventually EDNRB up-regulation. Under these conditions, EDN1 expression was still significantly increased relative to vehicle control, but nevertheless reduced when compared to BRAF inhibition alone (Fig 8I). This was intriguing as it suggested that BQ788, despite enabling MITF up-expression in the presence of BRAF inhibitor, had suppressed EDN1 expression within the tumours. One possible explanation is that BQ788 affects other potential EDN1 sources. Indeed, inhibiting EDNRB profoundly reduced the presence of endothelial cells (Fig EV6D and E) and fibroblasts (Fig EV6F and G), both of which were increased in response to BRAF inhibition (Fig EV6D and F).

If the paracrine action of EDN1 is crucial *in vivo*, reducing its expression should have an effect on AXL/EDNRA-expressing cells, even if BQ788 does not directly target EDNRA activity. We therefore assessed AXL expression in the tumours on various treatment regimes (Fig 8J and K). In untreated tumours, AXL expression was detectable in distinct areas, while other areas of the tumour were negative, confirming the idea of AXL heterogeneity *in vivo* (Fig 8K). Tumours that had resumed growth on BRAF inhibitor alone displayed a strong increase in AXL mRNA expression (Fig 8J). In these "progressed" tumours, nearly all cells stained positive for AXL, demonstrating that the number of AXL-high cells had increased (Fig 8K). However, the addition of each of the three EDNR inhibitors resulted in a profound reduction in AXL expression (Fig 8J and K). Intriguingly, the reduction in AXL-expressing cells occurred in patches and resulted in areas of AXL expression and areas of complete absence of AXL expression, which suggests no general suppression of AXL expression, but rather a loss of AXL-high cells.

Overall our data lead to a model, whereby in tumours on treatment with BRAF inhibitor-induced MITF up-expression in MITF-high cells increases intra-tumour EDN1 levels, which can act on MITF-high (paracrine or autocrine) as well as AXL-high cells to re-activate ERK (Fig 8L). While in MITF-high cells, BRAF-induced MAPK signalling is regulating proliferation and survival, partly through MITF (Wellbrock *et al*, 2008), in AXL-high cells it only

regulates cell cycle progression, whereby other signals are responsible for survival.

## Discussion

Within any tumour, there is a profound amount of innate variability, which includes genetically and phenotypically differing cancer cells and stromal cell make-up (De Sousa *et al*, 2013). This tumour heterogeneity is not only fuelling cancer progression, but is also majorly contributing to treatment failure.

Even within MITF-high melanomas MITF displays considerable heterogeneity in its expression, whereby cells expressing higher and lower levels of MITF are found within one tumour and located adjacent to each other (Chapman *et al*, 2014; Tirosh *et al*, 2016). This heterogeneity is critical for tumour progression as MITF expression levels define melanoma cell phenotypes of different proliferation rate and invasiveness, and we have shown recently that a co-operative communication between these MITF phenotypes can contribute to an overall increased invasiveness (Chapman *et al*, 2014). Intriguingly, MITF heterogeneity can be detected throughout progression and can even be found in circulating tumour cell clusters (Khoja *et al*, 2014), suggesting that maintaining heterogeneity throughout tumour progression is beneficial for the establishment of advanced disease.

Another marker for heterogeneity is AXL, which defines a population of cells with a more de-differentiated and invasive phenotype (Sensi *et al*, 2011; Muller *et al*, 2014). A recent elegant single-cell analysis revealed that all melanomas contain AXL-high populations to a certain degree and it is now well accepted that the AXL-high phenotype is linked to MAPKi therapy resistance and that there is an enrichment of the AXL-high phenotype in BRAFi/MEKi therapy post-relapse melanomas (O'Connell *et al*, 2013; Anastas *et al*, 2014; Muller *et al*, 2014; Ji *et al*, 2015; Tirosh *et al*, 2016).

There appears to be no difference in the frequency of AXL-high-resistant tumours that develop in patients who eventually relapse on BRAF inhibitor mono or BRAF/MEK inhibitor combination therapy. However, intriguingly, there might be a difference in the initial response to the different therapies. Exposure of MITF-high cell lines to MEK inhibitor or a BRAF/MEK inhibitor combination profoundly enriches for AXL-high cells within less than one week (Tirosh *et al*, 2016). While this could be due to reduced receptor shedding in response to MAPK pathway inhibition (Miller *et al*, 2016), enrichment for AXL-high cells is not a predominant initial reaction when MITF-high cell lines are treated with BRAF inhibitor alone (Fig EV1C–F). Tirosh *et al* reported similar results (Tirosh *et al*, 2016), suggesting that in the presence of BRAF inhibitor alone an instant selection for AXL-high subpopulations is outweighed by other mechanisms that enable individual MITF-high cells to survive the initial drug insult. We found that paracrine protection via EDNR signalling can be such a mechanism. Considering that paracrine protection appears to be not relevant when MEK is inhibited and the preferred option of BRAF/MEK inhibitor combination therapy in patients, addition of an EDNR antagonist would not be expected to create a benefit in this setting. However, our data suggest that a BRAF/EDNR inhibitor combination could be as potent with the advantage of not increasing AXL-high expression.

Recently, down-regulation of FRA1 (FOSL1) in response to MAPK inhibition was identified as common inducer of a paracrine acting secretome (Obenauf *et al*, 2015). Importantly, in melanoma FRA1 and MITF expression are mutually exclusive and low FRA1 expression indicates a cell state with high MITF expression (Muller *et al*, 2014; Verfaillie *et al*, 2015). This is entirely in line with our observation of increased MITF expression in response to MAPK inhibition and suggests that MITF is contributing to the MAPK inhibitor-induced secretome downstream of FRA1. Indeed, depletion of FRA1 from melanoma cells leads to up-regulation of MITF (Obenauf *et al*, 2015), but whether FRA1 directly acts as suppressor of *MITF* remains to be investigated.

Related to the "FRA1-induced" secretome IGF1 was found to be up-regulated within tumours in response to MAPK inhibition, and this IGF1 acted on innate-resistant cells supporting their outgrowth. We did not find IGF1 in our proteomics analysis; instead, we identified EDN1 as an essential factor within the MITF-induced secretome. Thereby, MITF not only regulated EDN1 expression, but also the expression of ECE1, the enzyme responsible for the production of the biologically active EDN1 peptide (Rossi *et al*, 2001). While we do not know whether EDN1 is a direct target gene of MITF, MITF has been identified at the *ECE1* promoter in ChIP-seq experiments (Strub *et al*, 2011). The EDN signalling pathway is closely linked to MITF and is crucial for the melanocytic lineage during development, but also in adult melanocytes (Saldana-Caboverde & Kos, 2010). This epistatic link is reflected in individual forms of Waardenburg syndrome (WS), an inherited sensorineural deafness condition, in which MITF mutations are implicated in WS type IIA and mutations in EDNRB (the relevant EDN receptor in melanocytic cells) in WS type IV (Saldana-Caboverde & Kos, 2010). Intriguingly, in the mouse Ednra is expressed in migrating neural crest cells (Clouthier *et al*, 1998), which are the de-differentiated precursors of melanocytes. This observation is entirely in line with the AXL-high phenotype of de-differentiation and increased motility and invasion (Sensi *et al*, 2011; O'Connell *et al*, 2013; Anastas *et al*, 2014; Konieczkowski *et al*, 2014; Muller *et al*, 2014; Ji *et al*, 2015; Tirosh *et al*, 2016). Thus, the neural crest origin, but de-differentiated nature of AXL-high melanoma cells might explain why they express preferentially EDNRA, while differentiation towards the melanocyte lineage, triggered by MITF expression, results in a switch to EDNRB expression. A similar distribution of receptors amongst phenotypes corresponding to MITF-high and AXL-high has been found with the WNT5A receptors ROR1 and ROR2, respectively (O'Connell *et al*, 2013). However, instead of maintaining both phenotypes, the presence of WNT5A results in down-regulation of ROR1 and induction of a phenotype switch (O'Connell *et al*, 2013). Thus, WNT5A rather contributes to the establishment of the AXL-high phenotype than the maintenance of heterogeneity.

Indeed, it appears crucial for the maintenance of heterogeneity that both phenotypes still respond to and are supported by EDN1. We show that while EDNR signalling protects MITF-high cells, it is also required for the AXL-high phenotype during treatment. This suggests that as long as the MAPK pathway is inhibited the increased EDN1 expression "nurtures" the otherwise under-represented population of AXL-high cells, which eventually can re-establish tumour growth in the presence of inhibitor even when pathway re-activation occurs and EDN1 levels drop again.

Our findings of EDN1 as unique regulator of phenotype heterogeneity maintenance and paracrine protection from BRAF inhibition add a novel feature to EDN1, which is known to be involved in many aspects of cancer development including EMT and chemotherapy resistance (Rosano *et al*, 2013). Interestingly, both EDNRB- and EDNRA-specific inhibitors had the potential to improve the response to BRAF inhibition, which appeared to be due to additional effects on the microenvironment, particularly endothelial cells and fibroblasts. EDN1 is well known to support tumour growth and progression through acting on the microenvironment (Rosano *et al*, 2013), and hence, interfering with EDNR signalling apart from targeting melanoma cell phenotypes could have the additional benefit of suppressing a favourable tumour microenvironment. Moreover, because the EDN1-induced paracrine signalling has the potential to support both MITF-high and AXL-high phenotypes in acquired resistant tumours, targeting the EDN1-EDN receptor axis could reduce the complexity seen in patients treated with MAPK inhibitor at time of progression.

# Materials and Methods

### Cell culture treatments and drug dose–response analysis

Melanoma cell lines were grown in DMEM/10% FCS. Cell numbers were measured as the optical density at 595 nm ($OD_{595}$) of solubilized crystal violet from formalin fixed cells. For all *in vitro* experiments, vemurafenib was used as BRAF inhibitor. Different MEK inhibitors (PD184352, selumetinib and trametinib) were used and are specified in the figure legends. For dose–response curves, cells were plated in 96-well plates and treated with serial dilutions of the indicated drugs. The $GI_{50}$ was calculated using GraphPad Prism version 6.00 (San Diego California, USA).

### Patient samples

Patients with mutant *BRAF*[V600]-positive metastatic melanoma were treated with either a BRAF inhibitor, or a combination of BRAF and MEK inhibitors (for patient characteristics, see Appendix Table S1). All patients were consented for tissue acquisition per an IRB-approved protocol (Office for Human Research Studies, Dana-Farber/Harvard Cancer Center). Tumour biopsies were obtained before treatment or at the indicated days of treatment.

### *In vivo* xenograft studies

All animal procedures involving animals were ethically approved by University of Manchester Animal Welfare and Ethical Review Bodies (AWERB) and carried out under licence in accordance with the UK Home Office Animals (Scientific Procedures) Act (1986), the guidelines of the Committee of the National Cancer Research Institute (Workman *et al*, 2010) and the University's Policy on the Use of Animals in Research. Animals were housed in the University of Manchester Biological Safety Unit. CD1® nude mice (female, 8 weeks of age) were injected s.c. with $4 \times 10^6$ A375 cells (in PBS). When animals had developed melanoma nodules of ~100 mm³, drug administration was initiated ($n$ = 5–6 mice per group). Treatment was by oral gavage once daily with vehicle

(5% DMSO, 95% water), or the respective drugs as indicated. After the indicated number of days, tumours were isolated and analysed as described. Zebrafish (*Danio rerio*) were raised and maintained at the University of Manchester Biological Services Unit. Zebrafish xenografts were generated by injection of approximately 1,000 melanoma cells in total (for details see Appendix Table S3) into the space surrounding the heart of embryos 48 h post-fertilization. Subsequently, groups of six larvae per condition randomly assigned were treated with either vemurafenib (200 nM) or the vehicle DMSO. The drug was added to the fish medium, and larvae were grown at 34°C ambient temperature in chorion water. Before drug addition (day 1) and 3 days after drug addition, anesthetized larvae were imaged using a Leica SP5 confocal microscope. Images were processed using Volocity software (Perkin Elmer, Cambridge, UK).

### Time-lapse FUCCI cell cycle analysis

To generate WM164 melanoma cell lines stably expressing the FUCCI constructs, mKO2-hCdt1 (30–120) and mAG-hGem (1–110) (Sakaue-Sawano *et al*, 2008) were subcloned into a replication-defective, self-inactivating lentiviral expression vector system and the lentivirus was produced by co-transfection of human embryonic kidney 293T cells. High-titre viral solutions for mKO2-hCdt1 (30/120) and mAG-hGem (1/110) were prepared and used for co-transduction into melanoma cell lines and subclones were generated by single-cell sorting (Haass *et al*, 2014). Cells were seeded in multiwell tissue culture plates, and time-lapse microscopy was performed using an Olympus IX-81 inverted fluorescence microscope equipped with an incubation chamber at 37°C and 5% $CO_2$. Images were taken at intervals of 15 min using 10× objective. Cells were treated with 160 nM dabrafenib, 80 nM trametinib or equivalent DMSO volumes. DRAQ7 dye (Beckman Coulter) was added to all conditions to a final concentration of 3 μM to track cell death. Cells were monitored and the occurrence of cell cycle phases as well as cell death recorded.

### Melanoma 3D spheroid growth

Melanoma cells were re-suspended in Dulbecco's modified Eagle's medium containing 5% foetal bovine serum and 0.32% methylcellulose (Sigma). The cell suspension was transferred into a 96-well plate (1,000 cells per well), and spheres were allowed to form over a period of 48–72 h as previously described (Ferguson *et al*, 2013). Spheres were then transferred into 0.5-ml fibrillar bovine dermal collagen (2.3 mg/ml; Nucaton, Leimuiden, The Netherlands) with one sphere per well of a 24-well plate. Once the collagen was set, Dulbecco's modified Eagle's medium containing 10% foetal bovine serum was added, and after approximately 16 h, drugs were added to the medium at indicated concentrations.

### Statistical analysis

If not indicated otherwise, data represent the results for assays performed in triplicate, with error bars to represent standard deviations or errors from the mean. Statistics used were as follows: predominately Student's *t*-test and one-way ANOVA with Tukey's *post hoc* test performed using GraphPad Prism version 6.0a for Mac OS,

**The paper explained**

**Problem**
Phenotype heterogeneity is a major challenge for targeted cancer therapy, and in melanoma, it can lead to the establishment of acquired resistant tumours with greater metastatic potential.

**Results**
We reveal that MITF driven communications between melanoma phenotype subpopulations establishes a paracrine protection to BRAF inhibitor therapy, allowing different phenotypes to prevail.

**Impact**
We show that targeting these communications by inhibiting EDNR signalling suppresses aggressive melanoma phenotypes and has the potential to improve the outcome of BRAF inhibitor therapy.

GraphPad Software, San Diego, California, USA, www.graphpad.com. Pearson correlation was used to analyse associated gene expression.

**Expanded View** for this article is available online.

## Acknowledgements

From the University of Manchester (UoM), we thank Dr. Helen Young and Dr. Adam Hurlstone for help with the zebrafish work, Dr. Brian Telfer for help with the mouse work, Peter Walker for help with the histology, and the 'UoM Strategic Fund' for the Histology Facility equipment. We thank Dr. Atsushi Miyawaki, RIKEN, Japan, for the FUCCI constructs, and Dr. Meenhard Herlyn, The Wistar Institute, Philadelphia, for the WM164 cells.  C.W. acknowledges support by Cancer Research UK (CRUK) [grant number C11591/A16416]; AICR/Worldwide Cancer Research [grant number 12-0235]; and by a Wellcome Trust institutional grant [grant number R116433]. The work was also supported by an NCI/NIH U54CA163125 grant to J.A.W. and K.T.F., and a K08CA160692 grant to J.A.W. N.K.H. is a Cameron fellow of the Melanoma and Skin Cancer Research Institute, Australia. N.K.H.'s contribution was supported by project grants RG 13-06 (Cancer Council New South Wales) and APP1084893 (National Health and Medical Research Council). The University Research Priority Program (URPP) in translational cancer research at the University of Zürich funded the generation of early passage melanoma cultures used in this work.

## Author contributions

Conception and design: MPS, NKH, CW; development of methodology: MPS, EJR, ZM, JF, LS, NKH, OS, SM, IA, AvK, JR, HB, JK; acquisition: MPL, RD, DTF, KTF, MCA, ZAC, JAW.

## Conflict of interest

Jennifer Wargo is a paid speaker for DAVA Oncology, Illumina and BMS and has served on advisory boards for Roche Genentech, GSK, and Novartis. All other authors declare no potential conflict of interest.

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
