## [Review Process File · EMBO Molecular Medicine]

Targeting endothelin receptor signaling overcomes heterogeneity driven therapy failure

Michael P. Smith, Emily J. Rowling, Zsofia Miskolczi, Jennifer Ferguson, Loredana Spoerri, Nikolas K. Haass, Olivia Sloss, Sophie McEntegart, Imanol Arozarena, Alex von Kriegsheim, Javier Rodriguez, Holly Brunton, Jivko Kmarashev, Mitchell P. Levesque, Reinhard Dummer, Dennie T. Frederick, Miles C. Andrews, Zachary A. Cooper, Keith T. Flaherty, Jennifer A. Wargo and Claudia Wellbrock

Corresponding author: Claudia Wellbrock, University of Manchester

Review timeline:

Submission date:	11 October 2016
Editorial Decision:	14 November 2016
Revision received:	31 March 2017
Editorial Decision:	27 April 2017
Revision received:	04 May 2017
Accepted:	05 May 2017

Transaction Report:

Editor: Roberto Buccione

1st Editorial Decision

14 November 2016

Thank you for the submission of your manuscript to EMBO Molecular Medicine. We have now heard back from the Reviewers whom we asked to evaluate your manuscript.

The three reviewers all find the manuscript to be quite interesting and of value for the community, but raise (especially reviewers 1 and 3) a number of very serious and in part overlapping concerns on the limited experimental support for many of the main claims including limited mechanist insight, which inevitably impacts on the overall clinical relevance of the findings. They would also like you to better explain the current findings vs. your previous work.

After reviewer cross-commenting and further discussion, it was agreed that the required experimentation is feasible and that therefore you should be allowed to revise your manuscript. In conclusion, while publication of the paper cannot be considered at this stage, we would be pleased to consider a suitably revised submission, provided, however, that the Reviewers' concerns are fully addressed with further experimentation where required and that acceptance of the manuscript will entail a second round of review

It is important that you consider that it is EMBO Molecular Medicine policy to allow a single round of revision only and that, therefore, acceptance or rejection of the manuscript will depend on the completeness of your responses included in the next, final version of the manuscript.

As you know, EMBO Molecular Medicine has a "scooping protection" policy, whereby similar findings that are published by others during review or revision are not a criterion for rejection. However, I do ask you to get in touch with us after three months if you have not completed your revision, to update us on the status. Please also contact us as soon as possible if similar work is published elsewhere.

Finally, please note that EMBO Molecular Medicine now requires a complete author checklist (<http://embomolmed.embopress.org/authorguide#editorial3>) to be submitted with all revised manuscripts. Provision of the author checklist is mandatory at revision stage; The checklist is designed to enhance and standardize reporting of key information in research papers and to support reanalysis and repetition of experiments by the community. The list covers key information for figure panels and captions and focuses on statistics, the reporting of reagents, animal models and human subject-derived data, as well as guidance to optimise data accessibility.

Please note that we now mandate that all corresponding authors list an ORCID digital identifier. You may do so through our web platform upon submission and the procedure takes <90 seconds to complete. We also encourage co-authors to supply an ORCID identifier, which will be linked to their name for unambiguous name identification.

I look forward to seeing a revised form of your manuscript as soon as possible.

***** Reviewer's comments *****

Referee #1 (Remarks):

In this article, Smith et al. investigated the effect of targeting endothelin receptor signaling to overcome resistance to BRAF inhibitor in melanoma. The current work provides new insight indicating that the paracrine ET-1 signaling controlled by MITF confers drug-resistance through ERK reactivation and that ET-1 receptor antagonist in combination with BRAFi can counteract essential process in melanoma drug resistance. Further studies should offer more definite insight into the specific mechanisms through which ET-1 receptor blockade might represent a possible novel therapeutic strategy for melanoma patients. These findings need to be strengthened by additional experiments.

1. The hypothesis that ET-1 gene is regulated by MITF in melanoma is intriguing, but the data are insufficient to offer more definite insight into the specific mechanism that regulate MITF/ET-1 axis. It is at transcriptional level?

2. By using the small cohort (n=11 patients, Table1) of Smith et al. 2016, the authors show that the expression level of ETBR is relatively similar than that of ETAR (Fig. 6G). Because the known role of the ETBR axis in melanoma development and progression, immunohistochemical analyses of targeted protein, such as ET1, ETAR, ETBR, and MITF, should be provided using larger cohort of melanoma patients.

3. Because in the previous report (Smith et al. 2016) the authors demonstrated that targeting MITF in the tolerance-phase could improve MAPK-inhibitor therapy response, the authors, beside the effect of bosentan in combination with BRAFi, should analyze the effect of the novel ET-1 receptor antagonist, macitentan, alone or in combination with BRAF and/or MAPK inhibitor, in melanoma xenografts, providing indication whether combination with macitentan can counteract melanoma MAPKi and/or BRAFi resistance. Similarly, the combination of BRAFi or MAPKi and macitentan should be evaluated on Erk and other downstream signaling.

4. ETBR is upregulated in most melanoma cell lines and is considered an indicator for melanoma progression, as its expression is enhanced in metastatic melanoma. ET-1 and ET-3 through ETBR enhance the survival of these cells and alter tumor-host interactions that lead to melanoma progression (J.Liu et al. 2014). Beside the paracrine role, to rule out the potential role of autocrine protective ETBR pathway in acquired resistant melanoma, the authors should perform experiments in cells silenced for ET-1 or ET-1 receptors or treated with macitentan.

5. To better define the role of ETBR or ETAR pathway as drivers of drug resistance the experiments of fig.5, 6 and 7 should be carried out with ETAR and/or ETBR specific antagonists, such as BQ123 or BQ788, respectively.

6. Moreover what happens in the experiment where "sensitive" melanoma cells are used and ET-1, ET-1R and/or MITF are overexpressed? Would these ET-1R and/or MITF overexpressing cells now be less sensitive to BRAFi or MAPKi? This experiment might add further support for targeting this signaling pathway in melanoma.

7. The acquisition of drug resistance can be explained in part by intrinsic properties of cancer cells, but could be dependent also by the tumor microenvironment. Indeed, targeting tumor cells and the microenvironment by using macitentan could represent a great potential for improving the prognosis of melanoma patients. In order to demonstrate that targeting ET-1 receptors can disable multiple signaling in melanoma cells and in the microenvironment, such as tumor-associated endothelial cells, the authors should investigate these effects in the melanoma xenograft tissues treated with ET-1R antagonist in mono and combination therapy. In addition, the heterogeneous expression of ET-1R on tumor cells and host components should be considered.

8. In order to evaluate whether combination treatment of macitentan with BRAFi (or MAPKi) could affect key mechanisms that are important for melanoma cell survival, the authors should analyze apoptosis.

9. ET-1-induced effects on AXL should be addressed with more mechanistic details. Moreover AXL and ETAR axis are poorly investigated. Further investigations are required to support the model depicted in Fig. 7F.

10. How combination of BRAFi and bosentan can promote the complete suppression of ETAR expression within the residual tumors (fig.6F) and the drop of AXL expression (Fig.6E)?

Minor comment: In the references the name of each journal should be abbreviated according to Index Medicus and italicized.

Referee #2 (Remarks):

In this manuscript, Smith and colleagues provide evidence that, upon exposure to a BRAF-inhibitor, BRAF-mutant melanomas upregulate expression of endothelin 1 (EDN1) in a MITF-dependent manner. EDN1 upregulation confers drug-resistance through ERK re-activation in a paracrine manner. Importantly, EDN1 support both MITF-high subpopulations through EDNRB and AXL-high cells through EDNRA and therefore promotes maintenance of melanoma heterogeneity during response to therapy. Endothelin receptor-antagonists suppress heterogeneity and sensitize to BRAF-inhibition.

This is an excellent and very interesting piece of work. The manuscript is well-written, solid and convincing.

The clinical relevance of the findings is, however, limited by the data presented figure S3D showing that the EDN1-paracrine protection can be overcome by MEKi. This is potentially a problem as BRAF/MEK combination therapy has replaced BRAF-inhibitor monotherapy in the clinic.

According to Figure S3D, EDN1 paracrine protection should be less efficient under combination BRAF/MEK inhibition. The authors should mention that this indeed limits the clinical relevance of their finding in their discussion.

Nevertheless, the findings are conceptually novel and important.

Below are my questions/comments/concerns:

-is EDN1-mediated paracrine protection only driven by MITFhigh cells? If so, what about the previous paper (Smith et al.) stating that nelfinavir leads a decrease in PAX3 driven MITF expression. Would MITFhigh driven EDN1-paracrine protection still occur?

- Introduction, Page3: "specific gene signature" correlating with AXL. The authors cite 5 papers and mention a specific signature. This is misleading. Even though these signatures have in common AXL they are not identical. The authors could propose a consensus signatures based on these 5 papers. Then the term "specific gene signature" would make sense.

- A375 cells might be genetically identical but transcriptionally they still may show heterogeneity (page5, A375 xenografts and MITF heterogeneity)
- Page 6 "30% of cells still displayed weak MITF expression". This is not clear from data presented in Fig.1D.
- Figure 2A: what are MITF and ERK basal levels before treatment?
- Figure 3B: Are data from cell lines M249, WM9 present in Figure1C? If so, it would be informative to indicate this.
- Figure 5D: why not include MEKi?
- Possible correlations between EDNRA, EDNRB, ECE1, MITF and AXL levels should not only be assessed in melanoma cell line but also in TCGA-SKCM (bulk) and in Tirosh's melanoma single cell RNAseq analysis.

Referee #3 (Remarks):

The manuscript by Smith et al. studies melanoma heterogeneity, which is reported to be maintained upon BRAF inhibitor treatment. Further investigating the link between melanoma heterogeneity and drug responses, the authors show that cells that are pre-treated with BRAF inhibitors support growth of BRAF inhibitor-sensitive tumor cells. This protective effect is associated with activation of the MAPK pathway and is mediated by endothelin-1. This secreted protein can protect against MAPK inhibitor-mediated loss of viability in sensitive cells via activation of the EDNRB receptor. Conversely, the authors provide evidence that in AXL-positive cells, which are generally less sensitive to MAPK inhibitors, endothelin-1 can provide protection via the EDNR receptors, too. This study concludes with *in vivo* evidence that targeting EDNR in combination with BRAF inhibition results in enhanced tumor killing.

I feel that this manuscript represents a study that is of potential interest to a broad audience and that it may provide a deeper understanding of how tumor heterogeneity can support melanoma survival in the context of MAPK inhibitor treatment. Nevertheless, a number of important concerns need to be addressed while some claims require better experimental evidence to provide sufficient backing for the paracrine support model that is put forward. Most importantly, it is unclear whether the paracrine effect is specific for cells that have been treated with MAPK inhibitor and that upregulate MITF (A375-T-like cells), or that something similar is observed for non-treated A375 cells too. To exclude this possibility, in Figures 2B, 3A, 5F, 7A and 7B, additional controls using (CM of) A375 cells should be included (see below for more specific comments). An important omission in the manuscript is that there is no support for the presence of A375-T-like cells *in vivo*. In the absence of this, Figure 6B does not support the model and might even contradict it (see below for more specific comments). Finally, the authors should further investigate what the supportive role of endothelin-1 is *in vivo* to provide a better mechanistic understanding on its function in paracrine growth support. In sum, while I believe that the message of the manuscript in principle is interesting indeed, my concerns are of fundamental nature and would require further investigation.

Major comments:

1) The authors insufficiently exclude the possibility that support by endothelin-1 is specific for cells that are A375-like (expressing higher levels of MITF upon MAPK inhibitor treatment). In a number of cases, the appropriate control is not included for this. For instance, in Figure 3A, CM from untreated cells should be included to show that the paracrine effect is dependent upon MAPK inhibition. In Figure 5F, the appropriate experiment would be to compare A375-T to A375 (rather than no) cells for co-injection with A375. With these experiments, it cannot be ruled out that it is the number of cells injected rather than what is injected that is important. Finally, in Figure 7B and 7C, CM from A375 cells should be included to show specificity of the paracrine effect for A375-T cells.

2) With respect to Figure 2B, it is unclear how this experiment was conducted. Can the authors exclude that the effect observed in the 'heterogeneous' setting is due to the fact they simply inject more cells? In other words, would injecting A375 + A375 (i.e., twice the amount) yield a different outcome than for A375 + A375-T? This again comes back to the question above as to how specific the 'paracrine effect' is for A375-T cells.

3) The authors make a point about the observation that in the absence of drug, A375 cells grow better in the context of A375-T cells. However, the reverse is also true, which is not explained in the

manuscript. This, once again, relates to my previous comments: can this be explained by the fact that more cells in total were injected, rather than that this is an effect of co-injection of the A375 and A375-T lines?

4) The authors should provide support for the presence of A375-T-like cells *in vivo*. While they show heterogeneous MITF expression in Figure 1, they fail to show that at least in some cells MITF is induced as a consequence of drug treatment. In addition to this, the potential dependence of MITF on the cell cycle should be taken into account when investigating this.

5) Related to point 4), if the authors cannot convincingly show that in Figure 6B A375-T-like cells (the ones upregulating MITF) are present, these experiments would not support the paracrine support model. Even worse, in such a case one could conclude that the effect of bosentan is aspecific, or that endothelin-1 secreted by regular (not A375-T-like) cells is sufficient to support tumor growth too, questioning the validity of the model.

6) The authors should provide a better understanding of the supportive role of endothelin-1 *in vivo*. How does this work? Is this via a decrease in apoptosis, a decrease in G1-arrest or an increase in proliferation rate?

7) With respect to Figure 3, a better characterization of the paracrine effect by the A375-T cells is required to properly assess it: i) how long is the paracrine effect observed after drug withdrawal; ii) is the paracrine effect correlated with MITF or AXL levels? Also, it is unclear where A375-T reside in the AXL-MITF expression landscape in Figure 1C. Since this is such an important cell line in this manuscript, it should be included there too.

Minor comments:

1) While the authors make the potentially interesting claim that "Intriguingly, we repeatedly observed that after the last round of cell division one daughter cell died, while the other daughter cell stayed arrested in G1 (Fig. 1E)", the referred experiment does not support this claim. Better experimental evidence, including a quantification of the phenomenon, is required to support this claim (which if true would be very interesting).

2) The description of a number of figure panels in the corresponding legends is missing. For instance, in Figure 1A, the (presumable) qRT-PCR is not mentioned in the legends. For Figure 1C, is the fold change in volume relative to day 1? For Figure 3A: which cells were treated with MAPKi to obtain CM?

3) In Figure 2A, expression of MITF in untreated cells should be included to assess the regulation of MITF by BRAFi treatment.

4) In Figure 3C, the color legend is very confusing (blue color generally denotes a decline in something, and that is not the case here).

5) With regards to Figure S5E and S5F, they provide important evidence that the paracrine effect is indeed mediated via endothelin-1. In my opinion this is key to the story and should be shown in the main figures.

6) It has been shown by several laboratories that acute MITF depletion in MITF-proficient tumors is lethal, and thus it is not surprising that MITF-depleted A375-T cells cannot support their A375 counterparts (Figure 4D).

7) By stating that "Together our data suggest that during the time of treatment when BRAF inhibition reduces MAPK pathway activity, both MITF/EDNRB-high and AXL/EDNRA-high populations are enriched. Indeed, we found that in melanomas from patients on treatment, both EDNRA and EDNRB expression was increased (Fig. 6G).", the authors imply that MITF/EDNRB-high and AXL/EDNRA-high populations are enriched. In order to support such a claim, the authors should exclude that BRAF inhibitor treatment does not simply upregulate EDNRA and EDNRB expression.

8) In Figure 7, the authors rely on bosentan as an inhibitor of EDNRs to show that these receptors support proliferation in AXL-high cells. However, to exclude that the effects observed here are specific, at least one independent way of blocking endothelin-1 signaling (e.g., using a blocking antibody or shRNA/CRISPR technology) should be included to support this claim.

9) In the summary the authors state that "Endothelin receptor-antagonists suppress heterogeneity", but I cannot find where this claim is supported in the manuscript.

1st Revision - authors' response

31 March 2017

***** Reviewer's comments *****

Referee #1 (Remarks):

In this article, Smith et al. investigated the effect of targeting endothelin receptor signaling to overcome resistance to BRAF inhibitor in melanoma. The current work provides new insight indicating that the paracrine ET-1 signaling controlled by MITF confers drug-resistance through ERK reactivation and that ET-1 receptor antagonist in combination with BRAFi can counteract essential process in melanoma drug resistance. Further studies should offer more definite insight into the specific mechanisms through which ET-1 receptor blockade might represent a possible novel therapeutic strategy for melanoma patients. These findings need to be strengthened by additional experiments.

1. The hypothesis that ET-1 gene is regulated by MITF in melanoma is intriguing, but the data are insufficient to offer more definite insight into the specific mechanism that regulate MITF/ET-1 axis. It is at transcriptional level?

In our manuscript we show that depletion of MITF results in reduced EDN1 mRNA expression (Figure 5B), indicating that the regulation of EDN1 downstream of MITF is at the transcriptional level. However, MITF also regulates the expression of ECE1 (Figure 4E), which is required for EDN1 processing. We now show that inhibiting ECE1 activity in melanoma cells reduces the amount of EDN1 detectable in the culture medium of these cells (new **Figure 4F**), suggesting that MITF also regulates the secretion of mature, active EDN1.

2. By using the small cohort (n=11 patients, Table1) of Smith et al. 2016, the authors show that the expression level of ETBR is relatively similar than that of ETAR (Fig. 6G).

The expression of EDNRA and EDNRB are not relatively similar; what was actually shown were the **fold changes on treatment**, which indeed are similar. In order to make clearer that in fact in melanoma EDNRA expression is much lower than EDNRB expression we are now showing the relative expression in **Figure 6F** and the relative fold change on treatment in patients in **Figure 6G**.

Because the known role of the ETBR axis in melanoma development and progression, immunohistochemical analyses of targeted protein, such as ET1, ETAR, ETBR, and MITF, should be provided using larger cohort of melanoma patients.

In our analysis of EDN1 and EDNRB expression we have included an additional number of 11 patient samples, which increases the number to 22, a reasonable cohort size we would argue. These data (shown in shown in **Figure 6A**) have been produced using qRT-PCR as we could not get hold of enough tissue samples for histology. However, we are providing more immunohistochemical analysis for MITF (an additional 5 patients), which confirms our findings with regard to heterogeneity. These data are shown in **Supplementary Figure S1**.

3. Because in the previous report (Smith et al. 2016) the authors demonstrated that targeting MITF in the tolerance-phase could improve MAPK-inhibitor therapy response, the authors, beside the effect of bosentan in combination with BRAFi, should analyze the effect of the novel ET-1 receptor antagonist, macitentan, alone or in combination with BRAF and/or MAPK inhibitor, in melanoma xenografts, providing indication whether combination with macitentan can counteract melanoma MAPKi and/or BRAFi resistance. Similarly, the combination of BRAFi or MAPKi and macitentan should be evaluated on Erk and other downstream signaling.

We have performed an *in vivo* experiment using macitentan alone or in combination with BRAF inhibitor. In **new Figure 8C** and **Figure EV6A** we show that it suppresses tumour growth, but is less efficient than bosentan or an EDNRB specific inhibitor, BQ788. We have also evaluated the effect of macitentan on ERK activity (DUSP6) and the downstream signalling that we had analysed for bosentan including the analysis for MITF, EDNRB, EDNRA and AXL expression. These data are shown in **new Figure 8**.

4. ETBR is upregulated in most melanoma cell lines and is considered an indicator for melanoma progression, as its expression is enhanced in metastatic melanoma. ET-1 and ET-3 through ETBR enhance the survival of these cells and alter tumor-host interactions that lead to melanoma progression (J.Liu et al. 2014). Beside the paracrine role, to rule out the potential role of autocrine protective ETBR pathway in acquired resistant melanoma, the authors should perform experiments in cells silenced for ET-1 or ET-1 receptors or treated with macitentan.

We don't want to rule out that EDN1 can act in an autocrine manner, *in vivo* this is in fact unavoidable and will definitely occur, and overall a balance of EDN1 signalling will be established within a tumour before and on treatment (we have included this now in our model in **new Figure 8L**). In support of this, it can be seen in Figure 5E that depleting the EDNRB receptor reduces cell growth (A375 vs A375 shEDNRB). The reviewer asks however about a potential role of autocrine EDNRB signalling in acquired resistant melanoma. In this case it will obviously depend on which mechanisms drives the acquired resistance, and as such this question does not have a single answer. We show for the reviewer (**Figure 1 for Reviewer#1**) that in cells that have acquired resistance due to NRAS overexpression (Nazarian et al, *Nature* 2010), EDNRB signalling is not a major contributor to cell growth, but in another acquired resistant cell line, in which the EGFR is overexpressed (Girotti et al, *Cancer Discov* 2013), inhibition with Macitentan reduces cell growth by 25%. Furthermore, if resistance is acquired through the outgrowth of AXL-high cells, then as we show in Figure 7C, EDN1 can rescue the growth inhibitory effect of BRAFi, but this occurs through EDNRA rather than through EDNRB (**new Figure 7D**).

Figure 1 for Reviewer#1:

A375/R cells over-express EGFR (Girotti et al, *Cancer Discov* 2013) and M249-AR4 cells are resistant to BRAFi due to NRAS overexpression (Nazarian et al, *Nature* 2010). Both cell lines were treated with macitentan either alone or in combination with BRAF inhibitor and relative cell number was analysed. Macitentan reduced the growth of both cell lines but only increased BRAF inhibitor response in A375/R cells.

5. To better define the role of ETBR or ETAR pathway as drivers of drug resistance the experiments of fig.5, 6 and 7 should be carried out with ETAR and/or ETBR specific antagonists, such as BQ123 or BQ788, respectively.

We have repeated these experiments using BQ788 and BQ123:

In Figure 5 the only drug experiment shown was a co-culture experiment in Fig 5G. This experiment using BQ788 and BQ123 is now shown in **new Figure EV5E**.

With regard to the *in vivo* experiment shown in Figure 6, we have used BQ788 *in vivo* and compared this with macitentan. We preferred macitentan over BQ123 as both drugs have a higher affinity to EDNRA than EDNRB, and as the reviewer already asked for an *in vivo* experiment using macitentan, we argued that we are addressing the general issue and do not perform a potentially

unnecessary animal experiment that would not have provided additional information. The new in vivo data are shown in **new Figure 8 and EV6**.

In Figure 7 we repeated the cell cycle progression and relative cell number experiments as well as a pERK Western blot, and the results are shown in **new Figure 7D and E, Figure EV5F and Figure S5A and B**.

6. Moreover what happens in the experiment where "sensitive" melanoma cells are used and ET-1, ET-1R and/or MITF are overexpressed? Would these ET-1R and/or MITF overexpressing cells now be less sensitive to BRAFi or MAPKi? This experiment might add further support for targeting this signaling pathway in melanoma.

We have ectopically overexpressed MITF in sensitive A375 cells, and this leads to increased EDN1 and EDNRB expression (**see new Supplementary Figure S4D**). This is in line with EDNRB being a MITF target gene (Sato-Jin K et al, FASEB 2008), and EDN1 a potential target gene. We and others have shown previously that overexpression of MITF results in resistance to BRAFi or MEKi (Smith et al, JNCI 2013, Muller et al, Nat Commun 2014), but we now also shown that cells ectopically overexpressing MITF can protect sensitive cells from BRAFi and that blocking EDNR signalling reduces ERK activity under these conditions (**new Supplementary Figure S4F and G**).

7. The acquisition of drug resistance can be explained in part by intrinsic properties of cancer cells, but could be dependent also by the tumor microenvironment. Indeed, targeting tumor cells and the microenvironment by using macitentan could represent a great potential for improving the prognosis of melanoma patients. In order to demonstrate that targeting ET-1 receptors can disable multiple signaling in melanoma cells and in the microenvironment, such as tumor-associated endothelial cells, the authors should investigate these effects in the melanoma xenograft tissues treated with ET-1R antagonist in mono and combination therapy. In addition, the heterogeneous expression of ET-1R on tumor cells and host components should be considered.

We thank the reviewer for this insightful comment. We have performed immunohistological staining for CD34 and aSMA, as well as qRT-PCR analyses for CD31 and aSMA (**see new Figure EV6D-G**). The results show that the different EDNR antagonists have different effects on stromal endothelial cells and fibroblasts. Our findings support and explain observations we make regarding for instance EDN1 levels within the tumours, and are described and discussed on page 17 of the manuscript.

8. In order to evaluate whether combination treatment of macitentan with BRAFi (or MAPKi) could affect key mechanisms that are important for melanoma cell survival, the authors should analyze apoptosis.

Further supporting our data shown in Figure 7D, we show that apoptosis occurs in A375 and WM98 cells (MITF-high/EDNRB-high), but not in WM793 and RPMI (AXL-high/EDNRB high) cells (**new Figure 7H**). We also show that in A375 xenografts apoptosis is increased in the various combination treatments (**new Figure 8D,E and EV6 B,C**).

9. ET-1-induced effects on AXL should be addressed with more mechanistic details. Moreover AXL and ETAR axis are poorly investigated. Further investigations are required to support the model depicted in Fig. 7F.

AXL expression is a marker for cells with a different transcriptional state in which RTKs are over-expressed (Muller et al, Nat Commun 2014). We show that EDN1 can reactivate ERK in the presence of BRAFi, and this is dependent on RAF and RTK signalling, as it can be blocked by RAF265 and the pan RTK inhibitor dovitinib. This is also reflected in cell growth, where we revealed that in contrast to signalling through EDNRB, this is independent of PKC (**see Figure EV5H and I**).

10. How combination of BRAFi and bosentan can promote the complete suppression of ETAR expression within the residual tumors (fig.6F) and the drop of AXL expression (Fig.6E)? We have updated our model in **new Figure 8L** and believe that the additional data we provide explain our model now in a better and clearer way. The histology provided in **new Figure 8K** also

should make clearer how AXL expressing cells appear to simply not increase in the presence of EDNR antagonists.

Minor comment: In the references the name of each journal should be abbreviated according to Index Medicus and italicized.

We have corrected this.

Referee #2 (Remarks):

In this manuscript, Smith and colleagues provide evidence that, upon exposure to a BRAF-inhibitor, BRAF-mutant melanomas upregulate expression of endothelin 1 (EDN1) in a MITF-dependent manner. EDN1 upregulation confers drug-resistance through ERK re-activation in a paracrine manner. Importantly, EDN1 support both MITF-high subpopulations through EDNRB and AXL-high cells through EDNRA and therefore promotes maintenance of melanoma heterogeneity during response to therapy. Endothelin receptor-antagonists suppress heterogeneity and sensitize to BRAF-inhibition.

This is an excellent and very interesting piece of work. The manuscript is well-written, solid and convincing. The clinical relevance of the findings is, however, limited by the data presented figure S3D showing that the EDN1-paracrine protection can be overcome by MEKi. This is potentially a problem as BRAF/MEK combination therapy has replaced BRAF-inhibitor monotherapy in the clinic. According to Figure S3D, EDN1 paracrine protection should be less efficient under combination BRAF/MEK inhibition. The authors should mention that this indeed limits the clinical relevance of their finding in their discussion. Nevertheless, the findings are conceptually novel and important.

We thank the reviewer for their positive comments. With regard to the clinical relevance, we discuss on page 19 that while BRAF inhibitor monotherapy might allow for paracrine signals and therefore does not enrich for AXL-high cells, addition of a MEK inhibitor would do so as selection is now favoured. We mention the limitations with regard to BRAF/MEK inhibitor combination therapy, but we suggest that an alternative to the combination therapy is BRAF/EDNR antagonist therapy which -as we demonstrate-suppresses high AXL expression.

Below are my questions/comments/concerns:

1 -is EDN1-mediated paracrine protection only driven by MITFhigh cells? If so, what about the previous paper (Smith et al.) stating that nelfinavir leads a decrease in PAX3 driven MITF expression. Would MITFhigh driven EDN1-paracrine protection still occur?

We show that in line with downregulation of MITF, EDN1 expression is reduced in xenografts from mice treated with a BRAFi Nelfinavir combination (**Figure 2 for Reviewer #2**). Therefore nelfinavir would counteract EDN1-mediated paracrine protection. Due to space limitation we have not included these data in the manuscript.

**Figure 2 for Reviewer#2:**

Combination of BRAFi and nelfinavir reduces EDN1 expression in A375 xenografts. Mice bearing A375 xenografts were treated with nelfinavir (25 mg/kg qd) or PLX4720 (BRAFi, 25 mg/kg qd) alone or in combination for 21 days, before RNA was extracted and analysed.

2- Introduction, Page3: "specific gene signature" correlating with AXL. The authors cite 5 papers and mention a specific signature. This is misleading. Even though these signatures have in common AXL they are not identical. The authors could propose a consensus signatures based on these 5 papers. Then the term "specific gene signature" would make sense.

We have analysed and identified a consensus signature (23 genes), but we believe that making this a point would rather distract from the focus at this stage of the manuscript. We have therefore removed the statement on page 3.

3- A375 cells might be genetically identical but transcriptionally they still may show heterogeneity (page5, A375 xenografts and MITF heterogeneity).

We agree with the reviewer, this is exactly the point we want to make, as different MITF transcription/protein levels will be linked to a different transcriptional state.

4- Page 6 "30% of cells still displayed weak MITF expression". This is not clear from data presented in Fig.1D.

We apologise for this, this was an estimated value based on what we saw in previous Figure S1 (now EV1E,F) and we have removed this statement.

5- Figure 2A: what are MITF and ERK basal levels before treatment?

We apologise for the confusion. The blot shown in Figure 2A is just depicting the basal MITF expression level in these cell lines in an untreated state. In order to not get confuse with the experiment regarding cell growth, we have separated these two results, and the MITF blot is now shown as **new Figure 2A** and the growth assay as **new Figure 2B**.

6- Figure 3B: Are data from cell lines M249, WM9 present in Figure1C? If so, it would be informative to indicate this.

We have done this.

7- Figure 5D: why not include MEKi?

We have done this. MEKi overcomes EDN1 mediated protection and this is now shown in **Figure EV4A**.

8- Possible correlations between EDNRA, EDNRB, ECE1, MITF and AXL levels should not only be assessed in melanoma cell line but also in TCGA-SKCM (bulk) and in Tirosh's melanoma single cell RNAseq analysis.

We have performed a correlation analysis using data from the TCGA dataset. The data confirm the correlative relationship of EDNRA and AXL and EDNRB and MITF we had observed in the cell line datasets. The data are shown in new **Figure 7A**. As for the Tirosh data, there is an uphold in the up-loading process. We have contacted the authors and this was their reply:

[MESSAGE OMITTED FOR PRIVACY REASONS]

Referee #3 (Remarks):

The manuscript by Smith et al. studies melanoma heterogeneity, which is reported to be maintained upon BRAF inhibitor treatment. Further investigating the link between melanoma heterogeneity and drug responses, the authors show that cells that are pre-treated with BRAF inhibitors support growth of BRAF inhibitor-sensitive tumor cells. This protective effect is associated with activation of the MAPK pathway and is mediated by endothelin-1. This secreted protein can protect against MAPK inhibitor-mediated loss of viability in sensitive cells via activation of the EDNRB receptor. Conversely, the authors provide evidence that in AXL-positive cells, which are generally less sensitive to MAPK inhibitors, endothelin-1 can provide protection via the EDNR receptors, too. This study concludes with *in vivo* evidence that targeting EDNR in combination with BRAF inhibition results in enhanced tumor killing.

I feel that this manuscript represents a study that is of potential interest to a broad audience and that it may provide a deeper understanding of how tumor heterogeneity can support melanoma survival in the context of MAPK inhibitor treatment. Nevertheless, a number of important concerns need to be addressed while some claims require better experimental evidence to provide sufficient backing for the paracrine support model that is put forward. Most importantly, it is unclear whether the paracrine effect is specific for cells that have been treated with MAPK inhibitor and that upregulate MITF (A375-T-like cells), or that something similar is observed for non-treated A375 cells too. To exclude this possibility, in Figures 2B, 3A, 5F, 7A and 7B, additional controls using (CM of) A375 cells should be included (see below for more specific comments). An important omission in the manuscript is that there is no support for the presence of A375-T-like cells *in vivo*. In the absence of this, Figure 6B does not support the model and might even contradict it (see below for more specific comments). Finally, the authors should further investigate what the supportive role of endothelin-1 is *in vivo* to provide a better mechanistic understanding on its function in paracrine growth support. In sum, while I believe that the message of the manuscript in principle is interesting indeed, my concerns are of fundamental nature and would require further investigation.

Major comments:

1) The authors insufficiently exclude the possibility that support by endothelin-1 is specific for cells that are A375-like (expressing higher levels of MITF upon MAPK inhibitor treatment). In a number of cases, the appropriate control is not included for this. For instance, in Figure 3A, CM from

untreated cells should be included to show that the paracrine effect is dependent upon MAPK inhibition.

Figure 3 for Reviewer #3 shows that A375 cells treated with medium from DMSO/untreated A375 cells do not develop tolerance to BRAF inhibition. Due to space limitations we did not include these data in the manuscript.

Figure 3 for Reviewer#3:

Dose response curves for BRAFi. A375 cells were treated in either DMEM (control) or the conditioned medium, which was derived from A375 cells treated with vemurafenib or DMSO for the indicated times.

In Figure 5F, the appropriate experiment would be to compare A375-T to A375 (rather than no) cells for co-injection with A375. With these experiments, it cannot be ruled out that it is the number of cells injected rather than what is injected that is important.

We apologise for this confusion as we realise that the labelling of this graph was not very helpful. We have improved the labelling to make clearer that in all zebrafish xenograft experiments the total number of injected cells is the same. We have also added a table demonstrating the composition of homogenous and heterogeneous xenografts in the zebrafish experiments in the Methods section in the Appendix.

Finally, in Figure 7B and 7C, CM from A375 cells should be included to show specificity of the paracrine effect for A375-T cells.

We have performed this control; the results are shown in **new Figures 7D and EV5F and G.**

2) With respect to Figure 2B, it is unclear how this experiment was conducted. Can the authors exclude that the effect observed in the 'heterogeneous' setting is due to the fact they simply inject more cells? In other words, would injecting A375 + A375 (i.e., twice the amount) yield a different outcome than for A375 + A375-T? This again comes back to the question above as to how specific the 'paracrine effect' is for A375-T cells.

We apologise for this confusion as we realise we did not explain this experiment very well. In all zebrafish xenograft experiments the total number of injected cells is the same, and we have made this now clearer in the figure legend. Furthermore, we have added a table demonstrating the composition of homogenous and heterogeneous xenografts in the zebrafish experiments in the Methods section in the Appendix.

3) The authors make a point about the observation that in the absence of drug, A375 cells grow better in the context of A375-T cells. However, the reverse is also true, which is not explained in the manuscript. This, once again, relates to my previous comments: can this be explained by the fact that more cells in total were injected, rather than that this is an effect of co-injection of the A375 and A375-T lines?

We consistently saw in our co-culture or conditioned medium experiments in-vitro that BRAFi-treated cells produce factors present in the medium that have a pro-proliferative effect. This is entirely in line with what we observed in vivo, and as explained above we inject the same total number of cells; we apologise again for not being clear enough about this.

4) The authors should provide support for the presence of A375-T-like cells *in vivo*. While they show heterogeneous MITF expression in Figure 1, they fail to show that at least in some cells MITF is induced as a consequence of drug treatment. In addition to this, the potential dependence of MITF on the cell cycle should be taken into account when investigating this.

We have previously demonstrated that A375-T cells exist *in vivo* (Smith et al, Cancer Cell 2916), but we now have added a blot in **Figure 1B** that shows MITF up-regulation in the ex-vivo cultures isolated from mice that had been treated with BRAFi. We did not emphasise on the increased expression level in the immunofluorescence experiment as the focus was on the heterogeneity, but also in this analysis MITF expression is increased.

5) Related to point 4), if the authors cannot convincingly show that in Figure 6B A375-T-like cells (the ones upregulating MITF) are present, these experiments would not support the paracrine support model. Even worse, in such a case one could conclude that the effect of bosentan is aspecific, or that endothelin-1 secreted by regular (not A375-T-like) cells is sufficient to support tumor growth too, questioning the validity of the model.

As mentioned above, we have previously demonstrated that A375-T cells exist *in vivo* (Smith et al, Cancer Cell 2916), and we now have added a blot in **Figure 1B** that shows MITF up-regulation in the ex-vivo cultures isolated from mice that had been treated with BRAFi. We also show that these ex-vivo cultures can produce paracrine protection (**new Figure EV3D**).

6) The authors should provide a better understanding of the supportive role of endothelin-1 *in vivo*. How does this work? Is this via a decrease in apoptosis, a decrease in G1-arrest or an increase in proliferation rate?

We have performed immunohistochemistry for cleaved caspase 3 and Ki67, and find that under conditions when EDN1 signalling is inhibited and tumour growth is reduced, Ki67 staining is also reduced and caspase staining is increased. This suggests that both, reduced proliferation and increased apoptosis are involved. These data are shown in **new Figure 8D and E**.

7) With respect to Figure 3, a better characterization of the paracrine effect by the A375-T cells is required to properly assess it: i) how long is the paracrine effect observed after drug withdrawal; ii) is the paracrine effect correlated with MITF or AXL levels? Also, it is unclear where A375-T reside in the AXL-MITF expression landscape in Figure 1C. Since this is such an important cell line in this manuscript, it should be included there too.

We have performed experiments showing that the paracrine effect reverts within 14 days (i) and that this correlates with MITF expression (ii). This is shown in **new Figure EV3B and C**. We have also added A375-T cells in Figure 1 C.

Minor comments:

1) While the authors make the potentially interesting claim that "Intriguingly, we repeatedly observed that after the last round of cell division one daughter cell died, while the other daughter cell stayed arrested in G1 (Fig. 1E)", the referred experiment does not support this claim. Better experimental evidence, including a quantification of the phenomenon, is required to support this claim (which if true would be very interesting).

We have now indicated these incidences by a dashed line and have toned down the description in the text as we do not want to distract from the actual point of the experiment, which was to demonstrate the heterogeneity in response and independence of MITF expression.

2) The description of a number of figure panels in the corresponding legends is missing. For instance, in Figure 1A, the (presumable) qRT-PCR is not mentioned in the legends. For Figure 1C, is the fold change in volume relative to day 1? For Figure 3A: which cells were treated with MAPKi to obtain CM?

We apologise for these omissions and have corrected this.

3) In Figure 2A, expression of MITF in untreated cells should be included to assess the regulation of MITF by BRAFi treatment.

We already showed in Figure 1B that long-term MAPKi inhibition up-regulates MITF expression, and Figure 2A was only meant to demonstrate the status quo of MITF expression in these cells. We do however realise that this was confusing and have now separated the blot and the growth experiment previously shown in Figure 2A, which are now **new Figure 2A and B**.

4) In Figure 3C, the color legend is very confusing (blue color generally denotes a decline in something, and that is not the case here).

We have changed this.

5) With regards to Figure S5E and S5F, they provide important evidence that the paracrine effect is indeed mediated via endothelin-1. In my opinion this is key to the story and should be shown in the main figures.

We agree with the reviewer about the relevance of these data. However, considering the amount of new data added to the manuscript and due to space limitations these are now in Figure EVB and C, where they will be more accessible than in a supplementary Figure.

6) It has been shown by several laboratories that acute MITF depletion in MITF-proficient tumors is lethal, and thus it is not surprising that MITF-depleted A375-T cells cannot support their A375 counterparts (Figure 4D).

MITF depletion does not induce death in all cell lines, and we show in **Figure 4 for Reviewer #3** that depleting MITF in A375-T cells. While reducing cell growth (due to cell cycle arrest), it does not induce cell death.

Figure 4 for Reviewer#3:

A. Relative cell number of A375 cells transfected with either a control or 2 MITF specific siRNAs. Cell number was analysed 48 h after transfection. B. Caspase activity in A375 cells transfected with either a control or 2 MITF specific siRNAs. Caspase activity using an incuCyte® was analysed 48 h after transfection. A BCL2 inhibitor was used as positive control.

7) By stating that "Together our data suggest that during the time of treatment when BRAF inhibition reduces MAPK pathway activity, both MITF/EDNRB-high and AXL/EDNRA-high populations are enriched. Indeed, we found that in melanomas from patients on treatment, both EDNRA and EDNRB expression was increased (Fig. 6G).", the authors imply that MITF/EDNRB-high and AXL/EDNRA-high populations are enriched. In order to support such a claim, the authors should exclude that BRAF inhibitor treatment does not simply upregulate EDNRA and EDNRB expression.

We agree with the reviewer and have removed this statement.

8) In Figure 7, the authors rely on bosentan as an inhibitor of EDNRs to show that these receptors support proliferation in AXL-high cells. However, to exclude that the effects observed here are specific, at least one independent way of blocking endothelin-1 signaling (e.g., using a blocking antibody or shRNA/CRISPR technology) should be included to support this claim.

We have added an experiment using macitentan, BQ788 and BQ123 as well as an EDN1 blocking antibody. The results are shown in **Figure EV5F and G**.

9) In the summary the authors state that "Endothelin receptor-antagonists suppress heterogeneity", but I cannot find where this claim is supported in the manuscript.

We agree with the reviewer that this statement does not accurately reflect what we wanted to say and we have changed the text, which now states that .."Endothelin receptor-antagonists suppress AXL-high expressing cells".

2nd Editorial Decision

27 April 2017

Thank you for the submission of your revised manuscript to EMBO Molecular Medicine. We have now received the enclosed reports from the referees that were asked to re-assess it. As you will see the reviewers are now globally supportive and I am pleased to inform you that we will be able to accept your manuscript pending the following final amendments:

- 1) Please provide scale bars for figures 6A, EV1C,D, E and F, EV2, EV6D and F. Also, scale bars for some EV6 panels are difficult to see, please improve visibility. Finally, please indicate in Fig. EV1B a scale bar for the magnification and indicate wherefrom it is derived in the original panel.
- 2) Although the signage on your source data files is quite good, it would be most helpful if you could also assign the corresponding panel identification in the source data files. In fact, I found it a bit difficult to properly match source blots with the corresponding manuscript figures. For instance I had such doubts for Fig. 3 and others. Please amend the source data files by indicating panel attribution but also please carefully check matches, especially for the loading controls.
- 3) While performing our pre-acceptance quality control and image screening routines, we also noticed possible lane splicing in Fig. 4. Furthermore, the corresponding source data file does not match the figure. Please note that as per our guidelines, this is not generally allowed. If however, juxtaposing images is essential, the borders should be clearly demarcated in the figure and declared/described in the legend. Please explain this occurrence, amend the figure and legend accordingly and provide the correct source data information.
- 4) Please move the "Time-lapse FUCCI cell cycle analysis", "Melanoma 3D Spheroid Growth, and "In vivo xenograft studies" sections of the Supplementary Methods to the main manuscript. Manuscript length is not a concern as EMBO Molecular Medicine is online only.
- 5) As per our Author Guidelines, the description of all reported data that includes statistical testing must state the name of the statistical test used to generate error bars and P values, the number (n) of independent experiments underlying each data point (not replicate measures of one sample), and the actual P value for each test (not merely 'significant' or 'P < 0.05').
- 6) The manuscript must include a statement in the Materials and Methods identifying the institutional and/or licensing committee approving the experiments, including any relevant details (like how many animals were used, of which gender, at what age, which strains, if genetically modified, on which background, housing details, etc). We encourage authors to follow the ARRIVE guidelines for reporting studies involving animals. Please see the EQUATOR website for details: <http://www.equator-network.org/reporting-guidelines/improving-bioscience-research-reporting-the-arrive-guidelines-for-reporting-animal-research/>. Please make sure that ALL the above details are reported. Age and gender details appear to be missing for the mouse experiments.
- 7) Every published paper includes a 'Synopsis' to further enhance discoverability. Synopses are displayed on the journal webpage and are freely accessible to all readers. They include a short description as well as 2-5 one-sentence bullet points that summarise the key NEW findings of the paper. The bullet points should be designed to be complementary to the abstract - i.e. not repeat the same text. We encourage inclusion of key acronyms and quantitative information. Please use the passive voice. Please attach this information in a separate file or send them by email, we will incorporate it accordingly. We also encourage the provision of striking image or visual abstract to illustrate your article. If you do, please provide a jpeg file 550 px-wide x 400-px high.

Please submit your revised manuscript within two weeks. I look forward to seeing a revised form of your manuscript as soon as possible.

***** Reviewer's comments *****

Referee #1 (Remarks):

The authors have responded appropriately to my concerns. I recommend acceptance for publication

Referee #2 (Remarks):

The authors have addressed adequately my concerns/criticisms.

Referee #3 (Remarks):

In my view, the revision of this manuscript has been performed in an outstanding manner, meticulously addressing all the questions and comments that I had. I can only compliment the authors for the precision with which they have revised their manuscript. As indicated in my previous review, I feel that this manuscript represents a study that is of interest to a broad audience and that will provide a deeper understanding of how tumor heterogeneity can support melanoma survival in the context of MAPK inhibitor treatment. I would therefore strongly recommend publication of the manuscript by Smith et al in EMBOMM.

2nd Revision - authors' response

04 May 2017

My sincere apologies for the mix-up with the source data; we have now addressed these issues (point 2), which were mainly 'flipped' images for the loading controls and the use of blots from other biological repeats of the respective experiments. We have now thoroughly checked that the source data match the images in the manuscript and have improved the labelling (point 2). As for Fig. 4 (point 3), I believe that the 'lane splicing' that was picked up was an issue with scanning the film. We have now provided the correct source and rescanned it to make sure that the 'line' is not appearing in the image.

Furthermore, we have added scale bars (point 1), have amended Materials and Methods (point 4), where we also added more information on the strain, age, gender and number of mice used in our study (point 6). We have also added exact P values and the number (n) of experiments/mice used for each data point (point 5). Finally, the labelling in the Appendix Figures has been corrected (extra email).

I have also written a Synopsis (point 7) that is accompanied by a visual abstract. I will be sending these files by email. I hope that we have addressed all the remaining issues and that the manuscript is now acceptable for publication.

Corresponding Author Name: Claudia Wellbrock

Journal Submitted to: EMBO MOLECULAR MEDICINE

Manuscript Number: EMM-2016-07156